# Iterative Feature Space Optimization through Incremental Adaptive Evaluation

## Abstract

Iterative feature space optimization involves systematically evaluating and adjusting the feature space to improve downstream task performance. However, existing works suffer from three key limitations: 1) overlooking differences among data samples leads to evaluation bias; 2) tailoring feature spaces to specific machine learning models results in overfitting and poor generalization; 3) requiring the evaluator to be retrained from scratch during each optimization iteration significantly reduces the overall efficiency of the optimization process. To bridge these gaps, we propose a gEneralized Adaptive feature Space Evaluator (EASE) to efficiently produce optimal and generalized feature spaces. This framework consists of two key components: Feature-Sample Subspace Generator and Contextual Attention Evaluator. The first component aims to decouple the information distribution within the feature space to mitigate evaluation bias. To achieve this, we first identify features most relevant to prediction tasks and samples most challenging for evaluation based on feedback from the subsequent evaluator. These identified feature and samples are then used to construct feature subspaces for next optimization iteration. This decoupling strategy makes the evaluator consistently target the most challenging aspects of the feature space. The second component intends to incrementally capture evolving patterns of the feature space for efficient evaluation. We propose a weighted-sharing multi-head attention mechanism to encode key characteristics of the feature space into an embedding vector for evaluation. Moreover, the evaluator is updated incrementally, retaining prior evaluation knowledge while incorporating new insights, as consecutive feature spaces during the optimization process share partial information. Extensive experiments on twelve real-world datasets demonstrate the effectiveness of the proposed framework. Our code and data are publicly available [1].

## 1 Introduction

Iterative feature space optimization systematically evaluates and refines the feature space to enhance downstream task performance (Jia et al., 2022). As depicted in Figure 1a, the optimization module iteratively enhances the feature space based on the feedback from the evaluator. This optimization process continues until the optimal feature space is identified. This approach has demonstrated broad applicability and has been successfully adopted in various fields, including biology, finance, and medicine (Zhu et al., 2023; Htun et al., 2023; Vommi & Battula, 2023).

Research in this domain has received significant attention (Zebari et al., 2020). Recursive optimization methods focus on evaluating feature importance to progressively refine the feature space (Darst et al., 2018; Priyatno et al., 2024). For instance, Escanilla et al. (2018) utilized sensitivity testing with membership queries on trained models to recursively identify key features. To improve the efficiency of these recursive methods, evolutionary algorithms and reinforcement learning (RL) were subsequently introduced, further accelerating the refinement process (Xiao et al., 2024; Wang et al., 2024; Liu et al., 2021a). For example, Wang et al. (2022) employed three cascading agents to replicate the feature engineering process typically performed by human experts, using RL to streamline the exploration phase.

---

[1] Https://anonymous.4open.science/r/EASE-1C51

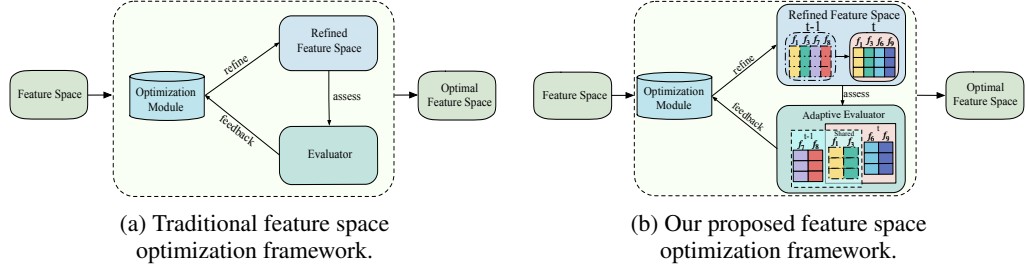

(a) Traditional feature space optimization framework.

(b) Our proposed feature space optimization framework.

Figure 1: (a) Illustration of the iterative feature space optimization, where the optimization module refines the feature space based on the feedback of the evaluator until the optimal one is identified. (b) The feature spaces between consecutive iterations exhibit informational overlap.

But, existing approaches suffer from three key limitations: **Limitation 1: Evaluation bias.** These methods do not account for variability between samples, which limits the evaluator's ability to capture the full range of features of the space. As a result, the performance assessments become biased and do not objectively reflect the quality of the feature space. **Limitation 2: Non-generalizability.** Customizing the feature space by replacing the evaluator based on specific requirements limits its ability to capture generalizable patterns. Consequently, the resulting feature space lacks flexibility and cannot be effectively applied across diverse scenarios. **Limitation 3: Training Inefficiency.** Retraining the evaluator from scratch at each iteration significantly increases computational demands. This technical trait leads to a time-consuming process that hinders efficiency and scalability.

Thus, there is a vital need for a robust evaluation framework that can efficiently assess feature space quality, enabling the creation of generalized and optimal feature spaces. This framework should integrate seamlessly with iterative feature space optimization algorithms to enhance their performance and efficiency. However, to accomplish this, there are two key technical challenges:

- **Challenge 1: Complicated Feature Interactions.** Within the feature space, complex feature-feature interactions exist, which are important for understanding its characteristics and enabling more effective refinement. But, how can we effectively capture such complicated information as the guidance during the iterative optimization process?

- **Challenge 2: Incremental Evaluator Updates.** As illustrated in Figure 1b, the feature spaces between consecutive iterations exhibit partial overlap. This overlap presents an opportunity to update the evaluator efficiently, rather than retraining it from scratch. But, how can we incrementally update the parameters of the evaluator to ensure it retains essential prior evaluation knowledge while simultaneously integrating new evaluation insights?

To address these challenges, we propose **EASE**, a g**E**neralized **A**daptive feature **S**pace **E**valuator, which can seamlessly integrate as a plugin into any iterative feature space optimization method for enhanced feature space refinement. This framework contains two key components: Feature-Sample Subspace Generator and Contextual Attention Evaluator. The first component aims to decouple the information distribution within the feature space to mitigate evaluation bias. To achieve this, we initially employ the feature index optimizer to select the features most relevant to the prediction task. Next, we use the sample index optimizer to identify the samples that present the greatest evaluation challenges. Both the previous two steps were guided by feedback from the subsequent evaluator. Finally, we use the identified features and samples to construct feature subspaces for the next iteration of feature space refinement. This decoupling strategy enables the evaluator to consistently target the most challenging aspects of the feature space, thereby facilitating the comprehensive comprehension and establishing a robust foundation for objective evaluation. The second component intends to incrementally capture the evolving patterns within the feature space for enhancing evaluation efficiency. Specifically, we employ a multi-head attention mechanism as the backbone to develop the evaluator. The feature subspaces are sequentially fed into the evaluator to capture complex relationships, leveraging contextual information across subspaces. The evaluator utilizes shared model weights across various feature subspaces. Moreover, since refined feature spaces across consecutive optimization iterations often share overlapping information, we incrementally update the evaluator's parameters to retain prior evaluation knowledge while incorporating new insights from the evolving feature space. Finally, we apply EASE to iterative feature selection algorithms and conduct extensive experiments on twelve real-world datasets to validate its superiority and effectiveness.

## 2 RELATED WORK

**Incremental Learning (IL).** IL aims to acquire new knowledge without forgetting the knowledge it has already learned (Zhu et al., 2021). IL is applied in scenarios such as dynamic environments (Shieh et al., 2020; Read et al., 2012) and online learning (Shim et al., 2021). IL methods can be divided into three categories: regularization, memory replay, and parameter isolation methods. Regularization-based methods (*e.g.*, Kirkpatrick et al. (2017); Li & Hoiem (2017)) prevent significant changes in important parameters of previous tasks. Memory replay methods retain old task data (Isele & Cosgun, 2018) or use generative models to simulate it (Shin et al., 2017), and then train this data alongside new task data when learning new tasks. Parameter isolation methods achieve task isolation by assigning independent model parameters to different tasks (*e.g.*, Rajasegaran et al. (2019); Serra et al. (2018)) or by expanding the network structure to accommodate new tasks (*e.g.*, Moriya et al. (2018); Aljundi et al. (2017)). In this paper, we update the evaluator using the Elastic Weight Consolidation (EWC) strategy (Kirkpatrick et al., 2017; Liu et al., 2021b). This approach estimates the importance of model parameters for previous tasks and minimizes changes to these important parameters when training on new tasks. This method significantly improves the training efficiency.

**Multi-Head Attention.** The multi-head attention mechanism enhances representation capability by simultaneously attending to different subspaces of the input data (Vaswani, 2017; Messaoud et al., 2021). This technique is used in natural language processing (Vaswani, 2017; Sun et al., 2020) and object detection (Dai et al., 2021) to capture complex patterns and dependencies. Unlike previous works, we propose a weighted multi-head attention mechanism that shares weights to encode key characteristics of the feature space into an embedding vector for the evaluation.

**Feature Selection (FS).** FS is widely used in high-dimensional fields (Nguyen et al., 2020), such as bioinformatics (Pudjihartono et al., 2022) and finance (Arora & Kaur, 2020). Among these FS methods, the wrapper method stands out for its ability to select features based directly on model performance (Nouri-Moghaddam et al., 2021). Wrapper-based methods use the performance of the downstream model as a criterion and employ iterative search to find the optimal feature subset (Liu et al., 2023). The most representative wrapper method is Recursive Feature Elimination (RFE). RFE iteratively trains the model and removes the least important features, gradually reducing the feature set size until a specified criterion is met (Guyon et al., 2002). In this paper, we use FS as a representative example of feature optimization to illustrate the subsequent technical details.

## 3 PROBLEM STATEMENT

This paper introduces a novel feature space evaluator to efficiently identify the optimal feature space. The proposed evaluator can be seamlessly integrated into any iterative feature space optimization algorithm. Formally, given a dataset $\mathbb{D} = \langle \boldsymbol{\mathcal{F}}, \boldsymbol{y} \rangle$, where $\boldsymbol{\mathcal{F}}$ represents the feature space and $\boldsymbol{y}$ denotes the target label space, we first initialize the parameters $\boldsymbol{\Theta}_{\mathcal{M}}$ of the evaluator $\mathcal{M}$ based on $\mathbb{D}$. This initialization is achieved by minimizing the prediction error $\mathcal{L}$. The learning objective can be defined as:

$$\arg \min_{\boldsymbol{\Theta}_{\mathcal{M}}} \mathcal{L}(\mathcal{M}(\boldsymbol{\mathcal{F}}; \boldsymbol{\Theta}_{\mathcal{M}}), \boldsymbol{y}). \tag{1}$$

In the $t$-th optimization, we can get a new feature space $\langle \boldsymbol{\mathcal{F}}^{(t)}, \boldsymbol{y}^{(t)} \rangle$. We leverage the information overlap between feature spaces from consecutive iterations to incrementally update $\boldsymbol{\Theta}_{\mathcal{M}}^{(t)}$, enabling efficient tracking of evolving patterns and providing an accurate evaluation of the feature space. The learning process can be formulated as follows:

$$\arg \min_{\boldsymbol{\Theta}_{\mathcal{M}}} \mathcal{L}(\mathcal{M}(\boldsymbol{\mathcal{F}}^{(t)}; \boldsymbol{\Theta}_{\mathcal{M}}^{(t)}), \boldsymbol{y}^{(t)}) + \lambda \|\boldsymbol{\Theta}_{\mathcal{M}}^{(t)} - \boldsymbol{\Theta}_{\mathcal{M}}^{(t-1)}\|_2, \tag{2}$$

where $\lambda$ is a regularization parameter that balances retaining prior evaluation knowledge with incorporating new insights from the updated feature space, and $\| \cdot \|_2$ is L2 norm. The learning process continues until either the maximum number of iterations is reached or the optimal feature space is identified. The design and optimization of $\mathcal{M}$ represent the core contribution of this paper. For clarity, key notations are summarized in Table 4 in the Appendix.

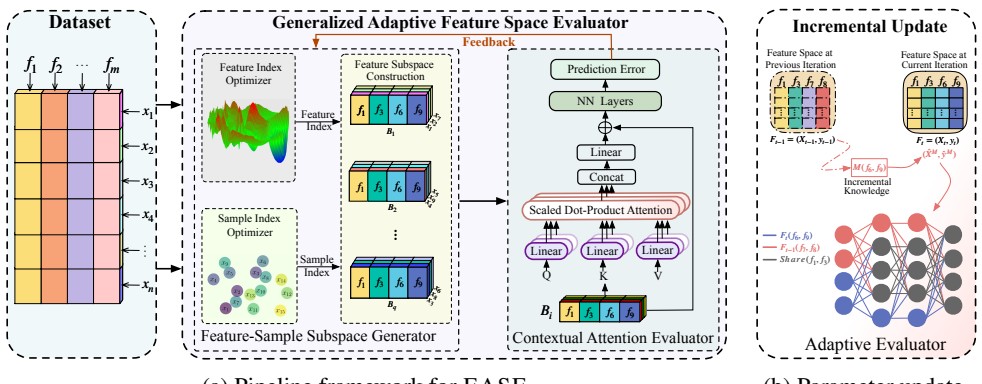

(a) Pipeline framework for EASE.          (b) Parameter update.

Figure 2: Framework overview and parameter update for EASE. The framework comprises two key components: the Feature-Sample Subspace Generator and the Contextual Attention Evaluator. The first component aims to decouple the complex information within the feature space, enabling the evaluator to focus on capturing the most challenging aspects for evaluation. The second component is designed to comprehensively capture the characteristics of the feature space, ensuring fair and accurate evaluation. By considering information overlap across consecutive iterations, the evaluator incrementally updates its parameters, enhancing the efficiency of the overall optimization process.

## 4 METHODOLOGY

**Framework Overview.** Figure 2a shows the framework overview of EASE, which includes two key components: 1) Feature-Sample Subspace Generator; and 2) Contextual Attention Evaluator. The first component decouples the information distribution within the evolving feature space, aiming to reduce the complexity for the subsequent evaluator in capturing the key characteristics of the feature space. Specifically, in each optimization, guided by the evaluator, we first use the feature index optimizer to identify the most important features for the downstream prediction task. Then, the sample index optimizer is used to discover samples that are challenging to evaluate. After that, we construct fixed-length feature subspaces by applying a random combination strategy to the identified features and samples. The second component aims to efficiently capture complex feature interactions within the feature space to facilitate effective evaluation. In detail, we first input the multiple feature subspaces constructed by the first component into the contextual attention evaluator to capture complex interactions and encode them into an embedding vector. In addition, the embedding vector is used to perform the evaluation task and the resulting prediction error is fed back to the previous component as guidance. During this process, as illustrated in Figure 2b, considering the partial information overlap between feature spaces in consecutive iterations, we incrementally update the evaluator's parameters to efficiently incorporate new evaluation insights.

### 4.1 FEATURE-SAMPLE SUBSPACE GENERATOR

**Why is feature space decoupling important?** During the iterative feature space optimization process, complex feature interactions can obscure the underlying patterns. A comprehensive understanding of these interactions is crucial for an accurate and objective evaluation. By decoupling the feature space, we can reduce the complexity of the learning task, allowing the evaluator to concentrate on the most challenging aspects, ultimately resulting in more effective and precise evaluations.

**Feature Index Optimizer** is designed to identify the features most relevant to the subsequent evaluation task. We derive the feature index subset based on the feature importance scores. Formally, in the $t$-th iteration, given the feature space $\boldsymbol{\mathcal{F}}^{(t)}$ and the target variable $\boldsymbol{y}^{(t)}$, the importance score $\text{Score}(\boldsymbol{f}_i)$ for each feature $\boldsymbol{f}_i$ is calculated by assessing the impact of removing that feature on the performance of the model. The importance score is computed as follows:

$$\text{Score}(\boldsymbol{f}_i) = \mathcal{L}(\mathcal{M}(\boldsymbol{\mathcal{F}}^{(t)}; \boldsymbol{\Theta}_{\mathcal{M}}^{(t)})) - \mathcal{L}(\mathcal{M}(\boldsymbol{\mathcal{F}}^{(t)} \setminus \{\boldsymbol{f}_i\}; \boldsymbol{\Theta}_{\mathcal{M}}^{(t)})). \tag{3}$$

Here, $\mathcal{L}(\mathcal{M}(\cdot;\boldsymbol{\Theta}_{\mathcal{M}}^{(t)})))$ denotes the feature space evaluator that measures model performance, and $\mathcal{F}^{(t)} \setminus \{\boldsymbol{f}_i\}$ represents the feature space $\mathcal{F}^{(t)}$ with feature $\boldsymbol{f}_i$ omitted. This means that we can estimate the $\text{Score}(\boldsymbol{f}_i)$ by removing $\boldsymbol{f}_i$ and observing the change in the loss. The loss function can be tailored to the specific task. For classification, cross-entropy loss is commonly used, whereas for regression, mean squared error is typically employed. Once the importance scores for all features have been computed, they are ranked in descending order. The features with higher ranks are considered to be the most significant contributors to the performance of the model. We select the best $k$ features based on their importance scores to identify the subset of the feature index $\boldsymbol{f}^{(t)} = \{\boldsymbol{f}_1, \boldsymbol{f}_2, \cdots, \boldsymbol{f}_k\}$ that is the most relevant to the evaluation task. This component can be replaced with any feature selection module, allowing EASE to be compatible with any iterative feature selection algorithm.

**Sample Index Optimizer** aims to select the most challenging samples for the subsequent evaluation task. The sample index subset is derived based on the evaluation error from the feature space evaluator. Formally, in the $t$-th iteration, the feature space consists of $n$ samples and a target variable $\boldsymbol{y}$. For the $i$-th sample $x_i$, the prediction error is given by $\mathcal{L}_i^{(t-1)} = \ell(\boldsymbol{y}_i, \hat{\boldsymbol{y}}_i)$, where $\ell$ denotes the evaluation metric, $\boldsymbol{y}_i$ is the target value, and $\hat{\boldsymbol{y}}_i$ is the prediction value. The sampling probability for sample $x_i$ is defined as: $P(X = x_i) = \frac{\mathcal{L}_i^{(t-1)}}{\sum_{j=1}^{n} \mathcal{L}_j^{(t-1)}}$, where $P(X = x_i)$ represents the likelihood of selecting sample $x_i$ based on its relative prediction error. This ensures that samples with large errors have a higher probability of being selected. To efficiently sample from this distribution, we use the cumulative distribution function (CDF), which allows us to transform a uniformly distributed random number into a sample from the desired probability distribution. The CDF is constructed as: $C_i = \sum_{k=1}^{i} p_k = \sum_{k=1}^{i} P(X = x_k)$, where $C_i$ represents the cumulative sum of probabilities up to the $i$-th sample. More specifically, in the weighted sampling process, we first generate a random number $r$ uniformly distributed in the interval $[0, 1]$. Next, we identify the first sample index $i$ such that the CDF satisfies $C_i \geq r$, and select the corresponding sample $x_i$. The process is repeated until a new set of sample indices $\mathbb{I}^{(t)}$ is collected. Using the CDF, the sampling process aligns with the distribution of prediction errors, giving higher priority to samples with larger errors.

**Feature Subspace Construction** decouples the complex feature space into distinct portions to improve the understanding of the subsequent evaluator for a fair evaluation. Thus, we introduce the strategy for the construction of feature subspaces using the set of feature index $\boldsymbol{f}^{(t)}$ and sample index $\mathbb{I}^{(t)}$. More specifically, we sample $s$ sample indices from $\mathbb{I}^{(t)}$ for $q$ times using repeated sampling to obtain various sub sample indices $\{\mathbb{I}_1^{(t)}, \mathbb{I}_2^{(t)}, \cdots, \mathbb{I}_q^{(t)}\}$ with the same length. Next, we use the obtained sample indices and $\boldsymbol{f}^{(t)}$ to select the corresponding samples and features, constructing various feature subspaces denoted as $\mathcal{B}^{(t)} = \{\boldsymbol{B}_1^{(t)}, \boldsymbol{B}_2^{(t)}, \cdots, \boldsymbol{B}_q^{(t)}\}$. The $i$-th feature subspace $\boldsymbol{B}_i^{(t)} \in \mathbb{R}^{s \times k}$, where $s$ is the number of data samples and $k$ is the number of features within $\boldsymbol{B}_i^{(t)}$. Through this process, we decouple the information distribution within the feature space, preserving the most important and challenging aspects for evaluation in $\mathcal{B}^{(t)}$. The pseudo-code for feature subspace construction is provided in Algorithm 1 in the Appendix for improving reproducibility.

## 4.2 CONTEXTUAL ATTENTION EVALUATOR

To thoroughly capture the complicated interactions of the feature space, we design a contextual attention evaluator leveraging a multi-attention mechanism. The learned feature subspaces $\mathcal{B}^{(t)}$, are sequentially fed into the evaluator to facilitate comprehensive information extraction. We use the $i$-th feature subspace to illustrate the following calculation process. For clarity, we omit the $(t)$ notation for the $i$-th example.

More specifically, we begin by projecting $\boldsymbol{B}_i$ into three different spaces: the query $\boldsymbol{Q}_i$, key $\boldsymbol{K}_i$, and value $\boldsymbol{V}_i$ spaces. These projections are computed through linear transformations, which can be defined as:

$$\boldsymbol{Q}_i = \boldsymbol{B}_i \cdot \boldsymbol{W}_Q, \boldsymbol{K}_i = \boldsymbol{B}_i \cdot \boldsymbol{W}_K, \boldsymbol{V}_i = \boldsymbol{B}_i \cdot \boldsymbol{W}_V, \tag{4}$$

where $\boldsymbol{W}_Q \in \mathbb{R}^{k \times d_k}$, $\boldsymbol{W}_K \in \mathbb{R}^{k \times d_k}$, and $\boldsymbol{W}_V \in \mathbb{R}^{k \times d_v}$ are learned weight matrices; $d_k$ is the dimensionality of the query and key spaces; $d_v$ denotes the dimensionality of the value space. Then, we compute the attention weights by taking the dot product between the query and key matrices. To ensure numerical stability and manageable gradient magnitudes during training, the result is scaled by $\sqrt{d_k}$ and normalized using the softmax function. These attention weights are used to perform a

weighted aggregation of the value matrix $\boldsymbol{V}_i \in \mathbb{R}^{s \times d_v}$, which can be formulated as:

$$\text{Attention}(\boldsymbol{Q}_i, \boldsymbol{K}_i, \boldsymbol{V}_i) = \text{softmax}\left(\frac{\boldsymbol{Q}_i \boldsymbol{K}_i^T}{\sqrt{d_k}}\right) \boldsymbol{V}_i. \tag{5}$$

To comprehensively capture multiple facets of the feature subspace, we design multiple heads, each with the same structure as described above. These heads generate different attention outputs from different perspectives. The resulting attention outputs are then concatenated and passed through a linear transformation to get $\boldsymbol{B}_i' \in \mathbb{R}^{s \times d_{\text{out}}}$, which can be formulated as

$$\boldsymbol{B}_i' = \text{concat}(\text{head}_1, \text{head}_2, \dots, \text{head}_h)\boldsymbol{W}_O, \tag{6}$$

where $h$ is the number of attention heads and $\boldsymbol{W}_O \in \mathbb{R}^{(h \cdot d_v) \times d_{\text{out}}}$ is the output weight matrix.

After that, we concatenate $\boldsymbol{B}_i'$ and $\boldsymbol{B}_i$ to form a combined representation, which is then passed through a fully connected layer to generate the prediction $\hat{\boldsymbol{y}}_i$. This process can be formulated as follows:

$$\hat{\boldsymbol{y}}_i = \text{FC}(\text{Concat}(\boldsymbol{B}_i'; \boldsymbol{B}_i)). \tag{7}$$

This concatenation allows the evaluator to retain both original and context enhanced feature information for more effective prediction. When different feature subspaces are input into the evaluator, the same structure is used, and the weights are shared across all subspaces. This ensures consistency and promotes generalization by learning common patterns.

### 4.3 OPTIMIZATION

**Pre-training.** A well-initialized contextual attention evaluator provides a strong foundation for evaluation, allowing faster convergence and ensuring fair evaluation. To ensure an effective initialization for the contextual attention evaluator $\mathcal{M}$, we pre-train it using the original feature space $\mathcal{F}$ as a foundational basis. Rather than employing the feature index optimizer and sample index optimizer, we construct the feature subspaces $\mathcal{B}^{(0)}$ by randomly sampling the feature and sample indices. Then, we subsequently input each feature subspace within $\mathcal{B}^{(0)}$ into the evaluator to perform the prediction. The optimization objective is to minimize the discrepancy between the predicted and actual values, which can be formulated as:

$$\mathcal{L}_{\text{intial}} = \sum_{i=1}^{s} \mathcal{L}_i(\boldsymbol{y}_i^{(0)}, \hat{\boldsymbol{y}}_i^{(0)}), \tag{8}$$

where $\boldsymbol{y}_i^{(0)}$ is the associated target label space of $\boldsymbol{B}_i^{(0)}$; $\hat{\boldsymbol{y}}_i^{(0)}$ is the predicted target label space; and $s$ is number of feature space within $\boldsymbol{B}_i^{(0)}$. After the model converges, the evaluator is initialized with the parameters $\Theta_{\mathcal{M}}^{(0)}$.

**Incremental Update.** In the iterative feature optimization framework, consecutive iterations often exhibit partial overlap in feature space information. This motivates us to incrementally update the parameters of the contextual attention evaluator $\mathcal{M}$, enabling faster updates and accelerating the entire feature space optimization process.

Specifically, in the $t$-th iteration, we begin by calculating the Fisher information to assess the importance of parameters based on the previous learning iteration. Given the feature subspaces $\mathcal{B}^{(t-1)} = \{\boldsymbol{B}_1^{(t-1)}, \cdots, \boldsymbol{B}_q^{(t-1)}\}$, the Fisher information for the $j$-th parameter $\boldsymbol{\theta}_j$ in the parameter set $\Theta_{\mathcal{M}}^{(t-1)}$ of the contextual attention evaluator is computed as:

$$\mathcal{G}(\boldsymbol{\theta}_j^{(t-1)}) = \frac{1}{s} \sum_{i=1}^{s} (\nabla_{\boldsymbol{\theta}_j} \log p(\boldsymbol{y}_i^{(t-1)} \mid \boldsymbol{B}_i^{(t-1)}; \Theta_{\mathcal{M}}^{(t-1)}))^2, \tag{9}$$

where $\mathcal{G}(\boldsymbol{\theta}_j^{(t-1)})$ measures the importance of $\boldsymbol{\theta}_j$ based on its contribution to the evaluation task in the $t-1$ iteration. The term $\nabla_{\boldsymbol{\theta}_j} \log p(\boldsymbol{y}_i^{(t-1)} \mid \boldsymbol{B}_i^{(t-1)}; \Theta_{\mathcal{M}}^{(t-1)})$ represents the logarithmic likelihood gradient with respect to $\boldsymbol{\theta}_j$ for conducting evaluation. A higher value of $\mathcal{G}(\boldsymbol{\theta}_j^{(t-1)})$ indicates that $\boldsymbol{\theta}_j$ is crucial to perform an evaluation of $\mathcal{B}^{(t-1)}$ (Grosse & Martens, 2016). After that, we impose constraints on parameter updates during the training of feature subspaces $\mathcal{B}^{(t)}$. The objective is to

prevent forgetting shared evaluation knowledge during parameter updates while incorporating new evaluation insights. The final loss function in the $t$-th iteration is defined as:

$$\mathcal{L}_{\text{final}}(\boldsymbol{\Theta}_{\mathcal{M}}^{(t)}) = \mathcal{L}_{\mathcal{B}^{(t)}}(\boldsymbol{\Theta}_{\mathcal{M}}^{(t)}) + \frac{\lambda}{2} \sum_j \mathcal{G}(\boldsymbol{\theta}_j^{(t-1)}) \left( \boldsymbol{\theta}_j^{(t)} - \boldsymbol{\theta}_j^{(t-1)} \right)^2, \tag{10}$$

where $\mathcal{L}_{\mathcal{B}^{(t)}}(\boldsymbol{\Theta}_{\mathcal{M}}^{(t)})$ represents the prediction loss for the current feature subspaces $\mathcal{B}^{(t)}$; $\boldsymbol{\theta}_j^{(t)}$ and $\boldsymbol{\theta}_j^{(t-1)}$ are the value of the $j$-th parameter from the parameter set of $\boldsymbol{\Theta}_{\mathcal{M}}^{(t)}$ and $\boldsymbol{\Theta}_{\mathcal{M}}^{(t-1)}$ respectively; $\lambda$ is a regularization factor that balances incorporating new evaluation knowledge with preserving shared knowledge. During the optimization procedure, we minimize $\mathcal{L}_{\text{final}}(\boldsymbol{\Theta}_{\mathcal{M}}^{(t)})$ to allow the contextual attention evaluator to efficiently capture the dynamics of the feature space, promoting faster convergence and more stable learning.

# 5 EXPERIMENTS

## 5.1 EXPERIMENTAL SETUP

**Datasets.** We conduct extensive experiments on 14 publicly available datasets from UCI (Public, 2024b), OpenML (Public, 2024c) and Kaggle (Public, 2024a), consisting of 6 classification tasks and 6 regression tasks. A statistical overview of these datasets is presented in Table 1. In this table, 'C' denotes dataset used for classification tasks and 'R' indicates datasets employed for represents regression tasks.

**Evaluation Metrics.** We use Mean Absolute Error (MAE), Root Mean Squared Error (RMSE), and R-squared ($R^2$) to evaluate the performance of regression tasks. Specifically, $R^2 = 1 - \frac{\sum_{i=1}^{n}(y_i - \hat{y}_i)^2}{\sum_{i=1}^{n}(y_i - \bar{y})^2}$, where $y_i$ and $\hat{y}_i$ respectively represent the true label and predicted label of $x_i$, and $n$ is the number of samples. We use Accuracy, Precision, Recall, and F1 score to evaluate the performance of classification tasks.

Table 1: Summary of the datasets.

| Dataset | R/C | Samples | Features | Classes | Source |
|---------|-----|---------|----------|---------|--------|
| openml_607 | R | 1000 | 51 | – | OpenML |
| openml_616 | R | 500 | 51 | – | OpenML |
| openml_620 | R | 1000 | 26 | – | OpenML |
| openml_586 | R | 1000 | 26 | – | OpenML |
| airfoil | R | 1503 | 6 | – | OpenML |
| bike_share | R | 10886 | 12 | – | OpenML |
| wine_red | C | 999 | 12 | 6 | UCI |
| svmguide3 | C | 1243 | 22 | 2 | OpenML |
| wine_white | C | 4898 | 12 | 7 | OpenML |
| spectf | C | 267 | 45 | 2 | UCI |
| spam_base | C | 4601 | 58 | 2 | OpenML |
| mammography | C | 11183 | 7 | 2 | OpenML |
| spam_base | C | 4601 | 58 | 2 | OpenML |
| AmazonEA | C | 32769 | 9 | 2 | Kaggle |
| Nomao | C | 34465 | 118 | 2 | UCI |

**Baseline Algorithms.** We apply EASE to two iterative feature selection frameworks to validate its effectiveness and generalization capability: (i) **RFE** (Guyon et al., 2002) iteratively eliminates the least important features from the original set until a stopping criterion is met. (ii) **FLSR** (Zhao et al., 2020) uses a single reinforced agent to perform feature selection with a restructured decision strategy. (iii) **SDAE** (Hassanieh & Chehade, 2024) is a state-of-the-art algorithm designed to select features used in unlabeled datasets without compromising information quality.

Additionally, we employ six widely-used ML algorithms as feature space evaluators during the iterative optimization process to compare their experimental performance against EASE: (1) Linear Regression / Logistic Regression (**LR**): Linear Regression (Su et al., 2012) models a linear relationship between the features and labels. Logistic Regression (Nusinovici et al., 2020) classifies data by linearly combining input features and applying a logistic function to the result. **LR** refers to linear regression for regression tasks and logistic regression for classification tasks. (2) Decision Tree (**DT**): (Kim & Upneja, 2014) is a tree-like structure method, used to classify or predict data through some rules. (3) Gradient Boosting Decision Tree (**GBDT**): (Li et al., 2023) builds an ensemble of decision trees sequentially to minimize errors. (4) Random Forest (**RF**): (Khajavi & Rastgoo, 2023) is an ensemble learning method that constructs multiple decision trees. (5) Extreme Gradient Boosting (**XGB**) (Asselman et al., 2023) combines the strengths of gradient boost with regularization techniques. In the testing phase, we use RF in all cases to report the performance of the refined feature space, as the model is stable and helps mitigate bias caused by the downstream model. For more experimental details, we have provided hyperparameters and environmental settings in Appendix C.1 to improve the reproducibility of our work.

Table 2: Overall performance comparison. The best results are highlighted in **bold**, and the second-best results are underlined. (↑ indicates that a higher value of the metric corresponds to better performance, while ↓ denotes that a lower value of the metric indicates better performance.)

| Dataset | R/C | Metrics | EASE | LR | DT | GBDT | RF | XGB |
|---|---|---|---|---|---|---|---|---|
| openml_607 | R | MAE↓ | **0.271** ± 0.028 | 0.342 ± 0.058 | 0.290 ± 0.020 | 0.277 ± 0.017 | 0.301 ± 0.025 | 0.278 ± 0.023 |
| | | RMSE↓ | **0.344** ± 0.036 | 0.439 ± 0.070 | 0.374 ± 0.019 | 0.352 ± 0.018 | 0.398 ± 0.041 | 0.345 ± 0.027 |
| | | $R^2$↑ | **0.873** ± 0.025 | 0.805 ± 0.087 | 0.847 ± 0.033 | 0.863 ± 0.010 | 0.835 ± 0.026 | 0.863 ± 0.026 |
| openml_616 | R | MAE↓ | **0.302** ± 0.039 | 0.315 ± 0.035 | 0.365 ± 0.011 | 0.321 ± 0.044 | 0.367 ± 0.037 | 0.323 ± 0.014 |
| | | RMSE↓ | **0.389** ± 0.048 | 0.404 ± 0.054 | 0.469 ± 0.023 | 0.418 ± 0.063 | 0.466 ± 0.038 | 0.406 ± 0.022 |
| | | $R^2$↑ | **0.840** ± 0.035 | 0.833 ± 0.028 | 0.799 ± 0.039 | 0.826 ± 0.044 | 0.791 ± 0.039 | 0.837 ± 0.016 |
| openml_620 | R | MAE↓ | **0.297** ± 0.017 | 0.371 ± 0.083 | 0.304 ± 0.014 | 0.302 ± 0.027 | 0.313 ± 0.015 | 0.298 ± 0.011 |
| | | RMSE↓ | **0.372** ± 0.021 | 0.476 ± 0.100 | 0.476 ± 0.100 | 0.380 ± 0.033 | 0.395 ± 0.016 | 0.383 ± 0.022 |
| | | $R^2$↑ | **0.861** ± 0.014 | 0.780 ± 0.072 | 0.852 ± 0.013 | 0.848 ± 0.031 | 0.846 ± 0.007 | 0.855 ± 0.021 |
| openml_586 | R | MAE↓ | **0.277** ± 0.011 | 0.313 ± 0.057 | 0.294 ± 0.024 | 0.291 ± 0.024 | 0.283 ± 0.020 | 0.281 ± 0.021 |
| | | RMSE↓ | **0.357** ± 0.019 | 0.405 ± 0.074 | 0.368 ± 0.033 | 0.373 ± 0.033 | 0.369 ± 0.025 | 0.361 ± 0.027 |
| | | $R^2$↑ | **0.875** ± 0.015 | 0.818 ± 0.062 | 0.862 ± 0.033 | 0.861 ± 0.022 | 0.862 ± 0.022 | 0.872 ± 0.010 |
| mammography | C | Accuracy↑ | **0.989** ± 0.003 | 0.978 ± 0.004 | 0.984 ± 0.004 | 0.987 ± 0.002 | 0.985 ± 0.005 | 0.985 ± 0.002 |
| | | Precision↑ | **0.947** ± 0.042 | 0.837 ± 0.036 | **0.976** ± 0.021 | 0.949 ± 0.018 | 0.947 ± 0.010 | 0.953 ± 0.024 |
| | | F1↑ | **0.818** ± 0.041 | 0.653 ± 0.035 | 0.713 ± 0.095 | 0.803 ± 0.041 | 0.757 ± 0.034 | 0.736 ± 0.028 |
| | | Recall↑ | **0.831** ± 0.043 | 0.603 ± 0.029 | 0.651 ± 0.078 | 0.733 ± 0.045 | 0.685 ± 0.035 | 0.662 ± 0.024 |
| spectf | C | Accuracy↑ | **0.825** ± 0.041 | 0.737 ± 0.028 | 0.781 ± 0.049 | 0.795 ± 0.055 | 0.815 ± 0.057 | 0.766 ± 0.033 |
| | | Precision↑ | 0.631 ± 0.248 | 0.467 ± 0.209 | **0.643** ± 0.237 | 0.466 ± 0.147 | 0.459 ± 0.117 | 0.482 ± 0.197 |
| | | F1↑ | **0.543** ± 0.054 | 0.454 ± 0.068 | 0.512 ± 0.077 | 0.478 ± 0.079 | 0.473 ± 0.058 | 0.450 ± 0.033 |
| | | Recall↑ | 0.515 ± 0.092 | 0.518 ± 0.036 | **0.542** ± 0.038 | 0.522 ± 0.044 | 0.514 ± 0.028 | 0.509 ± 0.018 |
| AmazonEA | C | Accuracy↑ | **0.963** ± 0.003 | 0.941 ± 0.004 | 0.944 ± 0.002 | 0.943 ± 0.002 | 0.944 ± 0.003 | 0.942 ± 0.003 |
| | | Precision↑ | 0.782 ± 0.199 | 0.670 ± 0.246 | 0.772 ± 0.245 | 0.489 ± 0.001 | **0.872** ± 0.199 | 0.611 ± 0.195 |
| | | F1↑ | **0.500** ± 0.001 | 0.488 ± 0.005 | 0.489 ± 0.003 | 0.490 ± 0.002 | 0.489 ± 0.003 | 0.488 ± 0.003 |
| | | Recall↑ | **0.503** ± 0.001 | 0.501 ± 0.002 | 0.502 ± 0.002 | 0.502 ± 0.001 | 0.502 ± 0.001 | 0.501 ± 0.002 |
| Nomao | C | Accuracy↑ | **0.952** ± 0.005 | 0.942 ± 0.002 | 0.944 ± 0.004 | 0.940 ± 0.003 | 0.941 ± 0.003 | 0.937 ± 0.001 |
| | | Precision↑ | **0.947** ± 0.004 | 0.940 ± 0.004 | 0.941 ± 0.003 | 0.937 ± 0.002 | 0.936 ± 0.003 | 0.935 ± 0.002 |
| | | F1↑ | **0.936** ± 0.007 | 0.927 ± 0.003 | 0.930 ± 0.005 | 0.924 ± 0.004 | 0.926 ± 0.004 | 0.922 ± 0.002 |
| | | Recall↑ | **0.922** ± 0.006 | 0.916 ± 0.003 | 0.920 ± 0.006 | 0.914 ± 0.006 | 0.917 ± 0.005 | 0.911 ± 0.002 |

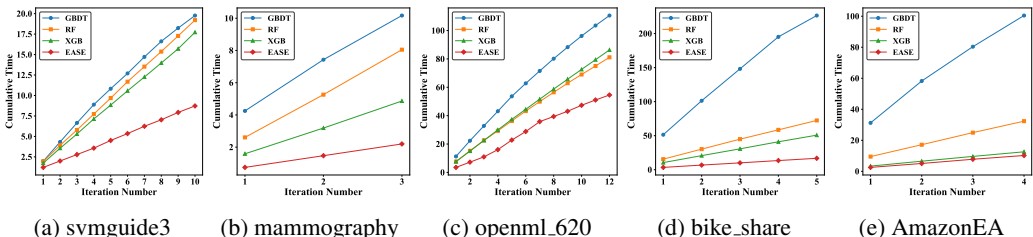

| (a) svmguide3 | (b) mammography | (c) openml_620 | (d) bike_share | (e) AmazonEA |
|---|---|---|---|---|

Figure 3: Time complexity comparison of different feature space evaluators across various datasets.

## 5.2 EXPERIMENTAL RESULTS

### 5.2.1 OVERALL COMPARISON

This experiment aims to answer: *Can* EASE *accurately assess feature space quality to produce an effective feature space?* We choose RFE for iterative FS and adopt EASE as the feature space evaluator. To compare the performance difference, we replace the evaluator with LR, DT, GBDT, RF and XGB respectively. We report the testing performance of the refined feature space using RF. Table 2 and Appendix C.2 shows the comparison results in terms of different evaluation metrics according to the task type. We find that EASE outperforms other baselines in most cases. For classification, EASE can improve by approximately 3% compared to other baselines. For regression, EASE demonstrates the most superior performance. The underlying driver is that our information decoupling strategy and context-aware evaluator, which allow the evaluator to focus on the most challenging aspects of the feature space. This results in a fairer evaluation, leading to a more effective refinement strategy and ultimately producing a more optimized feature space. In summary, this experiment shows that EASE effectively evaluates feature space quality for better feature spaces.

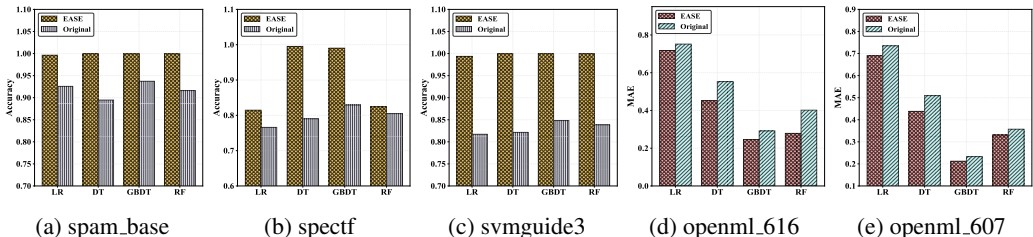

Figure 4: Comparison of prediction performance between original and EASE refined feature spaces.

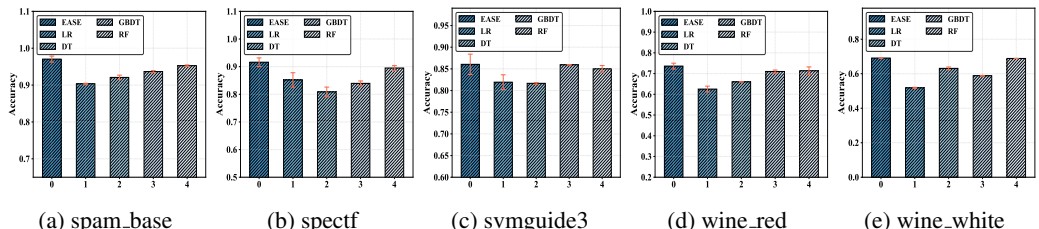

Figure 5: Comparison of refinement performance of feature space evaluators within FLSR.

### 5.2.2 EFFICIENCY COMPARISON

This experiment aims to answer: *Is EASE more efficient compared to other feature space evaluators?* We compare the training time of EASE with other feature space evaluators, including GBDT, RF, and XGB. Figure 3 shows the comparison results in terms of cumulative time. An interesting observation is that using EASE can significantly reduce the cumulative time costs compared to other baselines. A potential reason is that our incremental parameter update strategy enables the feature space evaluator to quickly capture evolving patterns in the feature space, thereby accelerating the feature optimization process. Additional experiments comparing time complexity across different datasets are provided in Figure C.3 in the Appendix. To sum up, this experiment demonstrates that EASE can efficiently assess feature space quality, thanks to its adaptive parameter update strategy.

### 5.2.3 THE EFFECTIVENESS OF EASE FOR FEATURE SPACE REFINEMENT

This experiment aims to answer: *Is the quality of the feature space refined by EASE superior to the original feature space?* We compare the prediction performance between the original feature space and the space refined by EASE using various downstream predictors, including LR, DT, GBDT, and RF. Figure 4 shows the comparison results in terms of Accuracy and MAE according to the task type. We find that the feature space produced by EASE outperforms the original in most cases across various predictors. In particular, the refinement by EASE outperforms the original feature space by 20% on the spectf dataset. This observation suggests that incorporating EASE into the iterative feature space optimization framework provides effective guidance for obtaining a better feature space. Additionally, the contextual attention evaluator comprehensively captures the intrinsic traits of the feature space, leading to robust performance across various datasets and predictors.

### 5.2.4 EASE'S PERFORMANCE IN DIFFERENT ITERATIVE FRAMEWORKS

This experiment aims to answer: *Is EASE generalizable and applicable across different iterative feature space optimization algorithms?* We apply EASE to a reinforced feature selection framework FLSR. To develop controlling groups, we replace the feature space evaluator within FLSR with LR, DT, GBDT, and RF, respectively. Figure 5 shows the comparison results across different datasets in terms of accuracy and performance standard deviation. We observe that EASE beats other baselines across various datasets. Especially, EASE improve the accuracy more than 5% in all situations. This observation highlights the strong generalizability and applicability of EASE. The underlying driver is that the feature index optimizer of EASE offers flexibility, allowing it to adapt to various iterative feature optimization frameworks. In summary, this experiment demonstrates that EASE exhibits

Table 3: Comparison of different EASE variants in terms of Accuracy or MAE. The best results are highlighted in **bold**, and the second-best results are underlined. ($\uparrow$ indicates that a higher value of the metric corresponds to better performance, while $\downarrow$ denotes that a lower value of the metric indicates better performance.)

| Dataset | R/C | Metric | EASE | EASE$^{-FC}$ | EASE$^{-IT}$ | EASE$^{-PT}$ |
|---|---|---|---|---|---|---|
| openml_607 | R | MAE$\downarrow$ | **0.271** $\pm$ 0.028 | 0.289 $\pm$ 0.028 | 0.343 $\pm$ 0.021 | 0.313 $\pm$ 0.035 |
| openml_616 | R | MAE$\downarrow$ | **0.302** $\pm$ 0.039 | 0.339 $\pm$ 0.035 | 0.435 $\pm$ 0.038 | 0.561 $\pm$ 0.082 |
| openml_620 | R | MAE$\downarrow$ | **0.297** $\pm$ 0.017 | 0.520 $\pm$ 0.018 | 0.383 $\pm$ 0.018 | 0.318 $\pm$ 0.017 |
| openml_586 | R | MAE$\downarrow$ | 0.277 $\pm$ 0.011 | 0.348 $\pm$ 0.016 | **0.276** $\pm$ 0.014 | 0.364 $\pm$ 0.005 |
| airfoil | R | MAE$\downarrow$ | **0.337** $\pm$ 0.009 | 0.751 $\pm$ 0.029 | 0.712 $\pm$ 0.028 | 0.362 $\pm$ 0.004 |
| bike_share | R | MAE$\downarrow$ | **0.033** $\pm$ 0.001 | 0.035 $\pm$ 0.002 | 0.034 $\pm$ 0.001 | 0.034 $\pm$ 0.001 |
| wine_red | C | Accuracy$\uparrow$ | **0.637** $\pm$ 0.018 | 0.589 $\pm$ 0.055 | 0.539 $\pm$ 0.043 | 0.539 $\pm$ 0.028 |
| svmguide3 | C | Accuracy$\uparrow$ | **0.846** $\pm$ 0.034 | 0.817 $\pm$ 0.032 | 0.828 $\pm$ 0.014 | 0.833 $\pm$ 0.010 |
| wine_white | C | Accuracy$\uparrow$ | **0.578** $\pm$ 0.017 | 0.557 $\pm$ 0.019 | 0.576 $\pm$ 0.014 | 0.552 $\pm$ 0.011 |
| spam_base | C | Accuracy$\uparrow$ | **0.939** $\pm$ 0.008 | 0.918 $\pm$ 0.016 | 0.928 $\pm$ 0.007 | 0.931 $\pm$ 0.003 |
| mammography | C | Accuracy$\uparrow$ | **0.989** $\pm$ 0.003 | 0.981 $\pm$ 0.004 | 0.985 $\pm$ 0.004 | 0.986 $\pm$ 0.003 |
| spectf | C | Accuracy$\uparrow$ | **0.825** $\pm$ 0.041 | 0.815 $\pm$ 0.065 | 0.756 $\pm$ 0.064 | 0.781 $\pm$ 0.041 |

strong adaptability to iterative feature space optimization frameworks and excellent generalizability across different optimization algorithms. We Additionally test the effectiveness of EASE and applied it to the state-of-the-art FS algorithm. The detailed results are provided in Appendix C.9.

### 5.2.5 THE IMPACT OF EACH TECHNICAL COMPONENT

This experiment aims to answer: *How does each technical component in* EASE *impact its performance?* We investigate the effects of pre-training, incremental training, and feature-sample subspace construction in EASE. We develop EASE$^{-PT}$, EASE$^{-IT}$, and EASE$^{-FC}$ by removing the pre-training, incremental training, and feature-sample subspace construction steps from EASE respectively. Table 3 shows the comparison results among different EASE variants. We observe that EASE outperforms EASE$^{-PT}$, highlighting the importance of the pre-training step in providing a strong foundation for objective evaluation. Additionally, EASE surpasses EASE$^{-IT}$, indicating that the incremental parameter updating mechanism effectively captures the evolving patterns of feature space optimization, leading to an improved feature space. Moreover, EASE outperforms EASE$^{-FC}$, showing that the information decoupling strategy reduces comprehension complexity, allowing for better capture of feature space characteristics and leading to improved evaluation and feature space quality. We also compare the time complexity of each component in Appendix C.6. In conclusion, this experiment reflects that each component in EASE is indispensable and significant.

For additional experiments and case studies, please refer to the Appendix, which further demonstrate the superiority, efficiency, and generalization capability of EASE.

## 6 CONCLUSION

In this paper, we propose a generalized adaptive feature space evaluator EASE for iterative feature space optimization. EASE consists of two key components: feature-sample subspace generator and contextual attention evaluator. The first component decouples the complex information within the feature space to generate diverse feature subspaces by the cooperation of the feature index optimizer and sample index optimizer. This enhances the ability of the subsequent evaluator to capture the most challenging information for more accurate feature space evaluation. The second component captures the intrinsic complexity of feature-sample interactions using a weight-sharing contextual-attention evaluator to ensure fair and accurate evaluation. Considering the information overlap across consecutive iterations, we incrementally update the evaluator's parameters to retain past knowledge while incorporating new insights. This allows the evaluator to efficiently capture the evolving patterns of the feature space. Extensive experimental results have demonstrated that EASE has achieved superior performance compared to other baselines. In addition, EASE exhibits strong adaptability, generalization, and robustness in various iterative feature optimization frameworks. In the future, we will focus on further enhancing the generalization capability of EASE to enable it to effectively handle distribution shifts and perform robustly across different types of datasets.

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

## A  TABLE OF NOTATIONS

We provide key notations in Table 4 to enhance the comprehension of the EASE methodology. Among these notations, $t$ represents the $t$-th optimization of the feature space algorithm.

Table 4: Notations for EASE.

| Notations | Interpretation |
| --- | --- |
| $\mathcal{M}^{(t)}$ | Model of EASE. |
| $\boldsymbol{\Theta}_{\mathcal{M}}^{(t)}$ | Parameter of EASE. |
| $\boldsymbol{\mathcal{F}}^{(t)}$ | Feature space. |
| $\mathcal{L}(\mathcal{M}(\cdot), \boldsymbol{y})$ | Prediction loss. |
| $\boldsymbol{f}^{(t)}$ | Feature index subset. |
| $\mathbb{I}^{(t)}$ | Sample index set. |
| $\mathcal{B}^{(t)}$ | Feature subspaces. |
| $\mathcal{G}_t(\boldsymbol{\theta}_j)$ | Fisher information for parameter $\boldsymbol{\theta}_j$. |
| $\mathcal{L}_{\text{final}}(\boldsymbol{\Theta}_{\mathcal{M}}^{(t)})$ | The final loss includes the incremental loss and prediction loss. |

## B  ALGORITHMS OF EASE

Algorithm 1 is the pseudo-code for feature subspace construction based on the feature space $\boldsymbol{\mathcal{F}}^{(t)}$ and the sample loss $\mathcal{L}_i^{(t-1)}$ from the previous iteration. Specifically, we first obtain an optimized feature index subset $\boldsymbol{f}^{(t)}$ by Feature Index Optimizer. Then, a sampling probability distribution $p_i$ and CDF $C_i$ are constructed. Next, $n$ samples are sampled to obtain the sample index set $\mathbb{I}^{(t)}$. Finally, the feature subspace $\mathcal{B}^{(t)}$ and $\boldsymbol{y}^{(t)}$ is constructed based on the $\mathbb{I}^{(t)}$, $s$ and $\boldsymbol{f}^{(t)}$.

---

**Algorithm 1** Feature Subspace Construction Algorithm.

---

1: **Input:** feature space $\boldsymbol{\mathcal{F}}^{(t)}$, fixed batch size $s$, sample loss $\mathcal{L}_i^{(t-1)}$.
2: **Output:** $\mathcal{B}^{(t)}$, $\boldsymbol{y}^{(t)}$.
3: $\boldsymbol{f}^{(t)} \leftarrow$ Feature index subset.
4: $p_i \leftarrow$ Sampling probability distribution.
5: $C_i \leftarrow$ CDF.
6: **for** $i = 0$ to $n$ **do**
7:     generate a random number $r \in [0, 1]$,
8:     find the first sample $x_i$ that $C_i \geq r$,
9:     add the sample index $i$ to the set $\mathbb{I}^{(t)}$.
10: **end for**
11: $\mathcal{B}^{(t)}, \boldsymbol{y}^{(t)} \leftarrow$ Base on $\boldsymbol{f}^{(t)}$, $s$ and $\mathbb{I}^{(t)}$.

---

Algorithm 2 is the pseudo-code for obtaining a robust evaluator. Specifically, we first construct a Contextual Attention Evaluator $\mathcal{M}$ and initialize its parameters $\boldsymbol{\Theta}$. Then, we use Algorithm 1 to obtain the feature subspace $\mathcal{B}^{(0)}$ of all features and to pre-train the $\mathcal{M}$. For the $t$-th evaluation, we obtain the feature subspace $\mathcal{B}^{(t)}$ at first, and then estimate the importance of each parameter $\mathcal{G}_{t-1}(\boldsymbol{\theta}_j)$, reducing the update magnitude for important parameters in previous iteration. Finally, we formulate the our objective function $\mathcal{L}_{\text{final}}(\boldsymbol{\Theta}_{\mathcal{M}}^{(t)})$ and minimize it.

---

**Algorithm 2** Generalized Adaptive Feature Space Evaluator Algorithm.

---

1: **Input:** Dataset $\mathbb{D} = \langle \mathcal{F}, \boldsymbol{y} \rangle$, optimization iterations $T$, fixed batch size s
2: **Output:** Evaluator $\mathcal{M}$.
3: $\mathcal{M} \leftarrow$ Contextual attention evaluator.
4: $\Theta \leftarrow$ Initialize the parameters.
5: $\mathcal{B}^{(0)}, \boldsymbol{y}^{(0)} \leftarrow$ Feature subspace construction.
6: $\Theta_{\mathcal{M}}^{(0)} \leftarrow$ Pre-training based on $\mathcal{B}^{(0)}$.
7: **for** $t = 0$ to $T$ **do**
8: $\quad \mathcal{B}^{(t)}, \boldsymbol{y}^{(t)} \leftarrow$ feature subspace construction,
9: $\quad \mathcal{G}_{t-1}(\boldsymbol{\theta}_j) \leftarrow$ estimate parameter importance,
10: $\quad \mathcal{L}_{\text{final}}(\Theta_{\mathcal{M}}^{(t)}) \leftarrow$ final loss .
11: $\quad \Theta_{\mathcal{M}}^{(t)} \leftarrow$ train $\mathcal{M}$ by minimizing $\mathcal{L}_{\text{final}}(\Theta_{\mathcal{M}}^{(t)})$.
12: **end for**

---

## C ADDITIONAL EXPERIMENT RESULTS

### C.1 EXPERIMENTAL SETUP

**Hyperparameters, Source Code and Reproducibility.** We limit pre-training and incremental training to 50 and 200, respectively. We employ an early stopping strategy, stopping the training process when the loss does not decrease for 10 consecutive epochs. In all experiments, we use the Adam optimizer and a learning rate decay strategy to accelerate the convergence. Specifically, the learning rate for the $t'$-th training iteration is:

$$l(t') = l(t'_0) \times p^{\left\lfloor \frac{t'}{u} \right\rfloor}, \tag{11}$$

where $l(t')$ and $l(t'_0)$ is current and initial learning rate, $p$ is the decay factor applied every $u$ iterations, and $\lfloor \cdot \rfloor$ is floor operation. And we set $l(t'_0) = 0.001$, $p = 0.9$ and $u = 30$. All experiments run 10 times and calculate the value of mean and standard deviation.

**Environmental Settings** All experiments were conducted on the macOS Sonoma 14.0 operating system, Apple M3 Chip with 8 cores (4 performance and 4 efficiency), and 8GB of RAM, with the framework of Python 3.8.19 and TensorFlow 2.13.0.

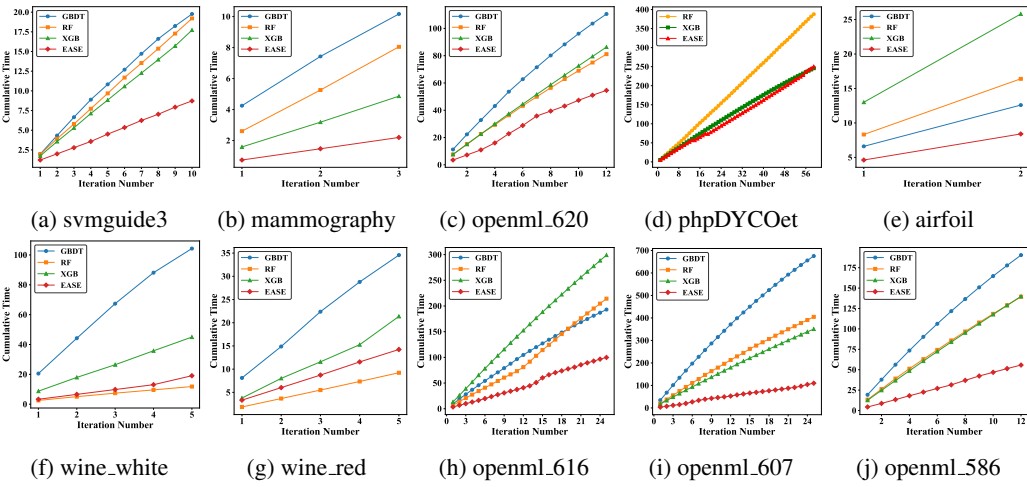

Figure 6: Time complexity comparison of different feature space evaluators across various datasets.

### C.2 ADDITIONAL OVERALL COMPARISON

We additionally compare overall comparison across all datasets. We choose RFE for iterative FS and adopt EASE as the feature space evaluator. To compare the performance difference, we replace the

Table 5: Overall performance comparison. The best results are highlighted in **bold**, and the second-best results are underlined. (↑ indicates that a higher value of the metric corresponds to better performance, while ↓ denotes that a lower value of the metric indicates better performance.)

| Dataset | R/C | Metrics | EASE | LR | DT | GBDT | RF | XGB |
|---|---|---|---|---|---|---|---|---|
| airfoil | R | MAE ↓ | **0.337** ± 0.010 | 0.348 ± 0.011 | 0.364 ± 0.014 | 0.339 ± 0.014 | 0.347 ± 0.017 | 0.347 ± 0.013 |
| | | RMSE ↓ | **0.440** ± 0.017 | 0.449 ± 0.014 | 0.475 ± 0.028 | 0.448 ± 0.016 | 0.454 ± 0.014 | 0.475 ± 0.030 |
| | | $R^2$ ↑ | **0.807** ± 0.010 | 0.801 ± 0.012 | 0.767 ± 0.012 | 0.799 ± 0.026 | 0.783 ± 0.013 | 0.802 ± 0.018 |
| bike_share | R | MAE ↓ | **0.033** ± 0.001 | 0.033 ± 0.002 | 0.034 ± 0.002 | 0.036 ± 0.000 | 0.034 ± 0.001 | 0.037 ± 0.001 |
| | | RMSE ↓ | **0.051** ± 0.002 | 0.053 ± 0.002 | 0.053 ± 0.003 | 0.057 ± 0.001 | 0.052 ± 0.001 | 0.052 ± 0.002 |
| | | $R^2$ ↑ | **0.998** ± 0.000 | 0.997 ± 0.000 | 0.997 ± 0.001 | 0.997 ± 0.001 | 0.997 ± 0.001 | 0.996 ± 0.000 |
| wine_red | C | Accuracy ↑ | **0.637** ± 0.018 | 0.617 ± 0.042 | 0.599 ± 0.043 | 0.596 ± 0.026 | 0.613 ± 0.020 | 0.588 ± 0.046 |
| | | Precision ↑ | **0.386** ± 0.062 | 0.311 ± 0.099 | 0.296 ± 0.018 | 0.264 ± 0.036 | 0.307 ± 0.066 | 0.285 ± 0.136 |
| | | F1 ↑ | **0.332** ± 0.057 | 0.260 ± 0.041 | 0.290 ± 0.017 | 0.263 ± 0.027 | 0.300 ± 0.045 | 0.269 ± 0.088 |
| | | Recall ↑ | **0.334** ± 0.060 | 0.276 ± 0.038 | 0.294 ± 0.015 | 0.276 ± 0.025 | 0.308 ± 0.031 | 0.283 ± 0.071 |
| svmguide3 | C | Accuracy ↑ | **0.846** ± 0.034 | 0.815 ± 0.015 | 0.816 ± 0.020 | 0.813 ± 0.024 | 0.817 ± 0.042 | 0.818 ± 0.050 |
| | | Precision ↑ | **0.852** ± 0.052 | 0.805 ± 0.033 | 0.823 ± 0.034 | 0.787 ± 0.044 | 0.823 ± 0.044 | 0.827 ± 0.065 |
| | | F1 ↑ | 0.646 ± 0.026 | 0.655 ± 0.048 | 0.673 ± 0.040 | 0.677 ± 0.059 | **0.691** ± 0.075 | 0.656 ± 0.074 |
| | | Recall ↑ | **0.679** ± 0.036 | 0.635 ± 0.038 | 0.651 ± 0.033 | 0.655 ± 0.047 | 0.670 ± 0.060 | 0.637 ± 0.055 |
| wine_white | C | Accuracy ↑ | **0.573** ± 0.017 | 0.531 ± 0.030 | 0.523 ± 0.027 | 0.549 ± 0.006 | 0.539 ± 0.026 | 0.546 ± 0.009 |
| | | Precision ↑ | **0.307** ± 0.074 | 0.259 ± 0.077 | 0.287 ± 0.114 | 0.303 ± 0.085 | 0.299 ± 0.089 | 0.273 ± 0.014 |
| | | F1 ↑ | 0.238 ± 0.016 | 0.214 ± 0.034 | 0.212 ± 0.037 | **0.240** ± 0.026 | 0.240 ± 0.030 | 0.218 ± 0.024 |
| | | Recall ↑ | **0.248** ± 0.018 | 0.223 ± 0.027 | 0.217 ± 0.028 | 0.244 ± 0.025 | 0.247 ± 0.025 | 0.230 ± 0.021 |
| spam_base | C | Accuracy ↑ | **0.939** ± 0.008 | 0.927 ± 0.017 | 0.929 ± 0.004 | 0.933 ± 0.009 | 0.936 ± 0.007 | 0.912 ± 0.012 |
| | | Precision ↑ | **0.939** ± 0.007 | 0.930 ± 0.017 | 0.932 ± 0.006 | 0.935 ± 0.009 | 0.939 ± 0.008 | 0.919 ± 0.010 |
| | | F1 ↑ | 0.929 ± 0.010 | 0.921 ± 0.018 | 0.925 ± 0.005 | 0.927 ± 0.009 | **0.932** ± 0.007 | 0.906 ± 0.011 |
| | | Recall ↑ | **0.928** ± 0.008 | 0.915 ± 0.019 | 0.919 ± 0.005 | 0.921 ± 0.009 | 0.927 ± 0.008 | 0.898 ± 0.011 |

evaluator with LR, DT, GBDT, RF and XGB respectively. Table 5 and Table 2 shows the comparison results in terms of overall performance. EASE as an evaluator, significantly enhances feature selection performance, demonstrating outstanding capability. This indicates that EASE not only possesses excellent generalization ability but also excels in evaluation performance. The potential reason for this superior performance lies in the Feature-Sample Subspace Generator, which greatly improves generalization, while the Contextual Attention Evaluator further optimizes performance by capturing interactions within the feature space.

## C.3 Additional Efficiency Comparison

We additionally compare training time across all datasets. We choose RFE for iterative feature selection and adopt EASE, GBDT, RF and XGB as the feature space evaluator respectively. Figure 6 shows the comparison results in terms of cumulative time. And we have omitted algorithms that consume too much time. We find that EASE outperforms other baselines nearly across all datasets. Specifically, EASE can save over 100 seconds of evaluation time compared to other time-consuming baselines on bike_share, wine_white, openml_616, openml_607, and openml_586. The underlying driver is that our incremental parameter update strategy focuses on the most relevant information and quickly capture evolving patterns in the feature space for evaluation, leading to a faster optimization process. In conclusion, EASE can effectively and efficiently evaluate the quality of the feature space.

## C.4 The Effectiveness of EASE for Feature Space Refinement

We additionally test the prediction performance on all datasets between the original feature space and the space refined by EASE using various downstream predictors, including LR, DT, GBDT, and RF. Figure 7 shows the overall comparison results in terms of Accuracy and MAE according to the task type. We find that the feature space evaluated by EASE outperforms the original feature space across all datasets and baselines. For the datasets openml_616, openml_607, spam_base, spectf, and svmguide3, and wine_white, EASE significantly improves performance in terms of Accuracy or MAE. The underlying driver is that our information decoupling strategy can effectively integrate information interactions and provide it to the contextual attention evaluator. Then contextual attention evaluator accurately captures the intrinsic interactions of feature space, thereby guiding the feature space iterative optimization algorithm to obtain high-quality feature space.

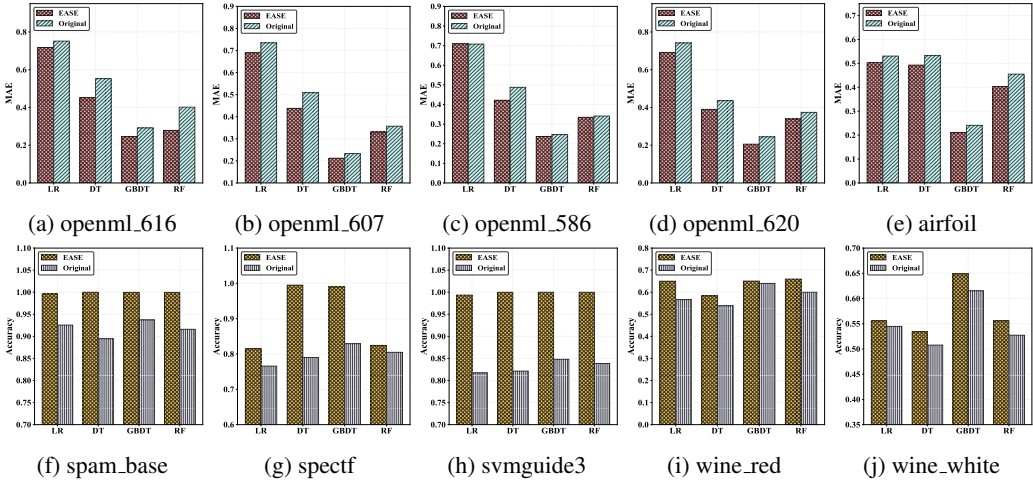

Figure 7: Comparison of prediction performance between original and EASE refined feature spaces.

## C.5 EASE'S PERFORMANCE IN DIFFERENT FEATURE SELECTION FRAMEWORKS

This experiment aims to answer: *Is* EASE *generalizable and applicable across different feature space optimization algorithms?* We apply EASE to a state-of-the-art FS algorithm SDAE (Hassanieh & Chehade, 2024). SDAE learns low-dimensional representations of high-dimensional data through a deep auto-encoder structure, while introducing a selective layer that automatically selects a relevant subset of features representing the entire feature space. This method performs FS in an unsupervised manner, effectively capturing nonlinear relationships between features. We respectively selected EASE LR, DT, GBDT, RF, and XGB as evaluators to assess the the final selected feature space. Table 6 shows the comparison results across different datasets. We can observe that in the feature space selected by the SDAE algorithm, EASE exhibits the best evaluation performance in both classification and regression tasks. This proves that EASE has stronger robustness and can accurately capture the key information in the feature space compared with other methods. The underlying reason for this lie in EASE's innovative design of the Contextual Attention mechanism.

Table 6: Comparison of different evaluators in terms of Accuracy (for classification tasks) and MAE (for regression tasks) in SDAE framework.The best results are highlighted in **bold**. The second-best results are highlighted in underline.(Lower MAE value and higher Accuracy value corresponds to better performance.

| Dataset | R/C | EASE | LR | DT | GBDT | RF | XGB |
|---|---|---|---|---|---|---|---|
| openml_607 | R | **0.232** ± 0.034 | 0.740 ± 0.040 | 0.397 ± 0.030 | 0.233 ± 0.015 | 0.278 ± 0.012 | 0.243 ± 0.011 |
| openml_616 | R | **0.202** ± 0.014 | 0.764 ± 0.045 | 0.518 ± 0.024 | 0.314 ± 0.026 | 0.343 ± 0.030 | 0.331 ± 0.033 |
| openml_620 | R | **0.199** ± 0.005 | 0.735 ± 0.033 | 0.477 ± 0.007 | 0.327 ± 0.013 | 0.358 ± 0.024 | 0.324 ± 0.012 |
| openml_586 | R | 0.228 ± 0.016 | 0.713 ± 0.041 | 0.383 ± 0.008 | **0.227** ± 0.019 | 0.264 ± 0.019 | 0.229 ± 0.010 |
| airfoil | R | **0.154** ± 0.013 | 0.564 ± 0.025 | 0.273 ± 0.012 | 0.286 ± 0.017 | 0.195 ± 0.008 | 0.162 ± 0.012 |
| bike_share | R | **0.007** ± 0.000 | 0.020 ± 0.000 | 0.016 ± 0.001 | 0.019 ± 0.000 | 0.007 ± 0.001 | 0.013 ± 0.001 |
| wine_red | C | **0.741** ± 0.037 | 0.591 ± 0.008 | 0.549 ± 0.022 | 0.640 ± 0.020 | 0.724 ± 0.032 | 0.600 ± 0.009 |
| svmguide3 | C | 0.856 ± 0.014 | 0.806 ± 0.021 | 0.795 ± 0.018 | 0.832 ± 0.024 | **0.861** ± 0.012 | 0.851 ± 0.004 |
| wine_white | C | **0.686** ± 0.034 | 0.518 ± 0.011 | 0.585 ± 0.010 | 0.584 ± 0.009 | 0.675 ± 0.015 | 0.654 ± 0.008 |
| spectf | C | 0.829 ± 0.046 | 0.789 ± 0.064 | 0.772 ± 0.030 | 0.756 ± 0.072 | **0.837** ± 0.057 | 0.813 ± 0.023 |
| mammography | C | **0.992** ± 0.001 | 0.983 ± 0.001 | 0.980 ± 0.003 | 0.986 ± 0.002 | 0.987 ± 0.004 | 0.987 ± 0.001 |
| spam_base | C | **0.963** ± 0.006 | 0.927 ± 0.007 | 0.904 ± 0.013 | 0.946 ± 0.003 | 0.960 ± 0.007 | 0.946 ± 0.005 |
| AmazonEA | C | **0.950** ± 0.003 | 0.944 ± 0.002 | 0.930 ± 0.003 | 0.942 ± 0.006 | 0.947 ± 0.002 | 0.943 ± 0.001 |
| Nomao | C | **0.973** ± 0.002 | 0.941 ± 0.004 | 0.945 ± 0.003 | 0.953 ± 0.002 | 0.967 ± 0.001 | 0.969 ± 0.002 |

## C.6 THE IMPACT OF EACH TECHNICAL COMPONENT

This experiment aims to answer: *How does each technical component in* EASE *impact its efficiency*? We compare the average training time of EASE with other EASE variants in each optimization, including $\text{EASE}^{-FC}$, $\text{EASE}^{-IT}$ and $\text{EASE}^{-PT}$. We develop $\text{EASE}^{-PT}$, $\text{EASE}^{-IT}$, and $\text{EASE}^{-FC}$ by removing the pre-training, incremental training, and feature-sample subspace

construction steps from EASE respectively. Table 7 shows the comparison results across different datasets in terms of average training time and standard deviation. We observe that EASE has the shortest runtime across all the datasets. Specifically, for classification, EASE$^{-IT}$ has the second-best performance which indicate the technical component of feature subspace construction and pre-training are crucial for enhancing efficiency. For regression, EASE$^{-PT}$ exhibit the second highest efficiency which demonstrate the technical component of incremental training and feature subspace construction can significantly improve EASE evaluation efficiency. The possible reason is pre-training can obtain a well-initialized contextual attention evaluator to provide a strong foundation for evaluation. Incremental training can leverage the overlapping information between consecutive iterations to avoid redundant computations and training time. And feature subspace construction can decouple the information within the feature space. In summary, our proposed EASE, which includes components for pre-training, incremental training, and feature subspace construction significantly reduce the average training time.

Table 7: Comparison of different EASE variants in terms of time complexity. The best results are highlighted in **bold**. The second-best results are highlighted in underline. (The unit is seconds.)

| Dataset | R/C | EASE | EASE$^{-FC}$ | EASE$^{-IT}$ | EASE$^{-PT}$ |
|---|---|---|---|---|---|
| openml_607 | R | **4.396** ± 1.237 | 5.400 ± 0.134 | 5.762 ± 0.044 | 5.175 ± 0.076 |
| openml_616 | R | **3.994** ± 1.347 | 5.408 ± 0.639 | 5.859 ± 0.051 | 5.097 ± 0.074 |
| openml_620 | R | **4.547** ± 1.322 | 5.635 ± 0.063 | 6.598 ± 0.068 | 5.687 ± 0.142 |
| openml_586 | R | **4.644** ± 0.382 | 7.287 ± 0.271 | 7.459 ± 0.056 | 6.980 ± 0.212 |
| airfoil | R | **4.201** ± 0.595 | 6.206 ± 0.340 | 6.531 ± 0.064 | 5.792 ± 0.340 |
| bike_share | R | **3.356** ± 0.069 | 5.877 ± 0.122 | 6.501 ± 0.116 | 5.745 ± 0.116 |
| wine_red | C | **2.848** ± 0.273 | 5.998 ± 1.006 | 3.235 ± 0.123 | 4.585 ± 0.062 |
| svmguide3 | C | **0.872** ± 0.134 | 5.899 ± 0.272 | 1.538 ± 0.468 | 6.178 ± 0.540 |
| wine_white | C | **3.815** ± 1.228 | 6.180 ± 0.225 | 4.083 ± 0.671 | 6.002 ± 0.103 |
| spam_base | C | **4.032** ± 0.354 | 7.598 ± 1.035 | 4.745 ± 0.829 | 7.622 ± 1.046 |
| mammography | C | **0.733** ± 0.010 | 1.436 ± 0.024 | 0.813 ± 0.088 | 4.799 ± 0.020 |
| spectf | C | **3.005** ± 0.043 | 5.852 ± 0.577 | 3.216 ± 0.301 | 5.276 ± 0.343 |

## C.7 PARAMETER SENSITIVITY ANALYSIS

This experiment aims to answer: *How do parameters affect the performance of* EASE*?* To validate the parameter sensitivity of key parameters in EASE, we select the wine_white and openml_586 datasets as examples. We focus on the number of heads $h$ and the embedding dimension $D$ in training procedure. To address the issue of varying feature space lengths during the iterative process, we first set the size of the feature subspace to match the embedding dimension and then transpose it, successfully overcoming this challenge. Consequently, $D$ is both the embedding dimension and the size of the feature subspace. Specifically, We set $D = 32$ and test the value of $h$ with the set $\{2, 4, 8, 16, 32\}$. And we set $h = 16$ and test the value of $D$ with the set $\{16, 32, 48, 64, 80, 96, 112, 128\}$. Figure 8 shows the comparison results in terms of Accuracy, Recall and F1 Score for classification task, 1-MAE, 1-RMSE, and $R^2$ Score for regression task. 1-MAE and 1-RMSE used for denoting that a higher value of the metric indicates better performance. We observe that the performance of downstream tasks generally remains stable across different values of $h$ and $D$, with significant changes occurring only at specific parameter values, such as $h = 4$ and $D = 64$ for the regression. A possible reason for this observation is that our proposed EASE can effectively decouple information within feature space and can capture contextual information during evaluation process. This observation indicates that EASE is not sensitive to the number of heads $h$ and the embedding dimension $D$. Therefore, the evaluating process of EASE is robust and stable.

## C.8 CASE STUDY

This experiment aims to answer: *What is the difference between the original feature space and the feature space refined by the* EASE *for ML tasks?* We select the wine_white dataset as example to visualize its features. In detail, we use the original feature space and a refined feature space evaluated by EASE within the RFE framework, with RF as the downstream predictor. Figure 9 shows the importance of top 8 features and their impact on the original feature space and EASE feature space. Specifically, we select 400 samples and calculate their contribution during the prediction process.

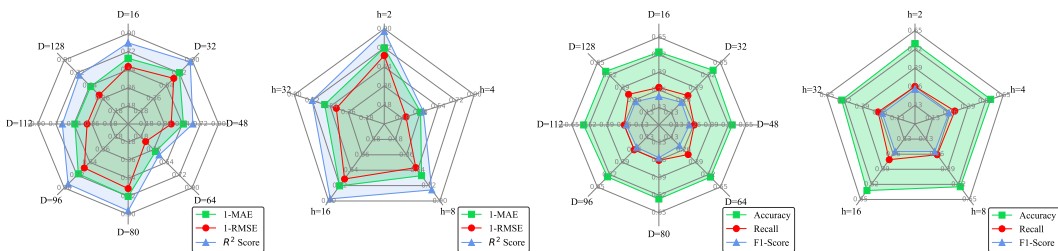

(a) $D$ on openml_586 (R).   (b) $h$ on openml_586 (R).   (c) $D$ on wine_white (C).   (d) $h$ on wine_white (C).

Figure 8: Parameter sensitivity on the number of heads $h$ and the embedding dimension $D$ on wine_white and openml_586 datasets.

Table 8: Comparison of different evaluators in terms of Accuracy (for classification tasks) and MAE (for regression tasks) in GRFG framework. The best results are highlighted in **bold**. The second-best results are highlighted in underline.

| Dataset | R/C | EASE | LR | DT | GBDT | RF | XGB |
|---|---|---|---|---|---|---|---|
| openml_616 | R | **0.302** ± 0.006 | 0.321 ± 0.002 | 0.327 ± 0.006 | 0.331 ± 0.012 | 0.312 ± 0.012 | 0.338 ± 0.000 |
| openml_586 | R | **0.373** ± 0.001 | 0.410 ± 0.008 | 0.410 ± 0.009 | 0.409 ± 0.003 | 0.385 ± 0.017 | 0.399 ± 0.002 |
| svmguide3 | C | **0.849** ± 0.002 | 0.816 ± 0.000 | 0.821 ± 0.006 | 0.822 ± 0.006 | 0.826 ± 0.000 | 0.820 ± 0.005 |
| mammography | C | **0.993** ± 0.001 | 0.984 ± 0.000 | 0.985 ± 0.018 | 0.986 ± 0.001 | 0.985 ± 0.000 | 0.986 ± 0.000 |

The horizontal axis represents the SHAP values for each feature, reflecting the impact of that feature on the prediction, while the vertical axis lists the feature names in order of importance (Temenos et al., 2023). And we color the size of the feature values (red represents larger values, while blue represents smaller values). We find that the EASE feature space greatly enhances the predictor accuracy by 15%. Another interesting observation is that the feature ranking in the EASE feature space differs from that in the original feature space for the same predictor RF. In detail, we can trace and explain the source and effect of specific feature. For example, "volatile acidity" measures the impact of the wine's acidity on the wine quality evaluation, which is positively correlated with wine quality. The underlying driver for these observations is that the multi-head attention mechanism in contextual attention evaluator can capture the interactions between samples and features after decoupling the information, which not only improves the fairness of the evaluation but also enhances its interpretability. Thus, this case study reflects that the effectiveness and interpretability of EASE as a evaluator for feature space quality evaluation.

For all other details of the hyperparameter configurations, optimization strategies, specific training processes, and environmental settings, please refer to Appendix C.1.

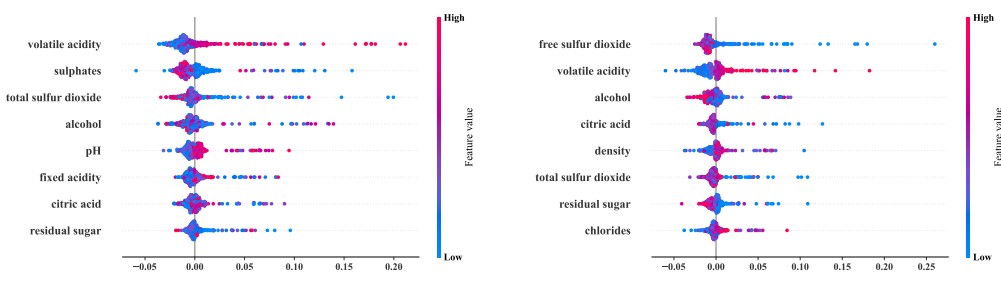

(a) EASE Feature Space (Accuracy = 0.5730).   (b) Original Feature Space (Accuracy = 0.4215).

Figure 9: Comparison of feature importance in EASE feature space and original feature space.

## C.9 EASE's Performance in Feature Generation Frameworks

This experiment aims to answer: *Is EASE generalizable and applicable in feature generation algorithm?* We apply EASE to a feature generation method GRFG (Wang et al., 2022). GRFG

addresses challenges in representation space reconstruction by proposing a cascading deep reinforcement learning approach that automates feature generation through a group-wise strategy and nested interactive processes. We respectively selected EASE LR, DT, GBDT, RF, and XGB as evaluators to evaluate the performance of the generated or selected features during the GRFG procedure. Table 8 shows the comparison results across different datasets. We observe that the proposed EASE achieves the best performance in both classification and regression tasks. This further demonstrates that, compared to traditional feature evaluation algorithms, EASE exhibits excellent performance in both feature selection and feature generation tasks and can capture the key information of the feature space.

