# OpenReview forum: "Iterative Feature Space Optimization through Incremental Adaptive Evaluation"
_ICLR.cc/2025/Conference — Submitted to ICLR 2025_

### Official Review · Reviewer_WB9F · 2024-10-17

**Soundness:** 2
**Presentation:** 1
**Contribution:** 2
**Rating:** 5
**Confidence:** 3

**Summary:**

This paper proposes a method to iteratively find good feature space (specifically, a set of relevant features in the original features) to improve downstream task performance. For this, the authors propose a Feature-sample subspace generator that finds relevant features and difficult samples in each optimization step and a Contextual attention evaluator that evaluates the selected features so that it can improve the prediction performance. The experiments show that the proposed method outperformed existing iterative feature space optimization methods.

**Strengths:**

- The feature selection is an important task in machine learning.
- The effectiveness of the proposed method was evaluated with many datasets including classification and regression tasks and many evaluation metrics.

**Weaknesses:**

- The significance of the task of iterative feature space optimization is not clear. For example, even if you don't find a feature space (specifically, feature selection) iteratively, lasso or neural network-based embedding-based feature selection methods can select features that directly improve the performance of downstream tasks by optimizing a single objective function once. This paper may be trying to solve unnecessarily complex problems.
- The presentation quality and clarity of this paper are low. For example, in Eq. 1, the evaluator $M$ takes the feature space as input and calculates the loss with the label space. What is the loss between spaces? Isn't it correct to say data (feature) matrix and label vector, etc., rather than spaces? The word 'space' is associated with mathematical vector spaces, etc., which can be confusing. I also need help understanding Eq. 3. Is the score a scalar or a vector? Because the specific form of $F^t$ (whether it's a vector, a matrix, or something else) is not described, the output of $M$ is also unclear. Thus, the definition of Eq. 3 is ambiguous.
Also, Eq. 2 in section 3 is different from Eq. 10, which is actually used. What is the intention of introducing a specific formula Eq. 2 at the problem-setting stage? I think that this makes the paper unnecessarily difficult to understand.I think the overall explanation of the proposed method needs to be improved.

**Questions:**

- Since this paper is considering feature selection, wouldn't it be easier for readers to understand if the authors propose the method as a feature selection method rather than using the abstract term 'feature space optimization'?
- In Figure 3, why is the proposed method faster than other comparison methods? Since the proposed method is based on neural networks (attention), I think it is more expensive.

---

> ### Author Response · Authors · 2024-11-17
>
> Dear reviewer WB9F：
>
> Thank you very much for taking the time to thoroughly review our manuscript. Your valuable comments have provided us with many useful insights and directions for improvement. We have carefully read all the questions and suggestions and will address them one by one.
>
> 1.Regarding significance of the task of iterative feature space optimization
>
> Indeed, existing traditional algorithms, such as Lasso, can perform feature selection in a single step. However, many iterative feature selection or feature generation algorithms can address the limitations of one-time selection or generation methods by capturing the complex interactions between features to improve performance. This approach has been mentioned in several recent works, as cited in the related literature in our introduction. The purpose of our proposed EASE framework is to provide a general evaluation method for these types of algorithms.
>
> 2.Regarding some expressions
>
> (1)Regarding Equation (1): The detailed derivation of the loss between two consecutive iterations is explained in Section 4.1, "Sample Index Optimizer." In the t-th iteration, we use the sample loss from the previous iteration, so this sample evaluation spans across two iterations. Since this part mainly aims to explain the methodology, we do not provide a detailed explanation of the specific loss calculation here. In Section 4.3, we further clarify the specific loss function we used.
>
> (2)Regarding terminology: In machine learning tasks, it is indeed common to refer to features as the data matrix. However, in this paper, we refer to the input data as the "feature space" to align with the common terminology used in feature selection, feature generation, and other feature optimization tasks. Specifically, "feature space optimization" is more specific than "data matrix optimization." If we used terms like "data matrix optimization" or "obtaining the data matrix," it could cause confusion with feature selection or feature generation tasks. Therefore, we chose to use the term "feature space" in this context.
>
> (3)Regarding Equation (3): In Section 3 of this paper, we provide a detailed definition of F and also mention the definition of F_t. Since F_t is a feature space, it is clear that F_t is a matrix. To avoid redundancy, we did not redefine symbols that were already defined earlier in subsequent equations. The score is a scalar, and we may not have explained this part sufficiently, which could have led to a lack of clarity for the readers. Therefore, we have revised the relevant content in the latest version.
>
> (4)Regarding Equations (2) and (10): Equation (2) is intended to clarify the incremental update method we use in the problem statement, specifically explaining that our incremental update strategy minimizes the empirical risk while constraining changes to some important model parameters. Equation (10), on the other hand, combines the feature subspace we construct to show how to constrain changes in model parameters, particularly through the use of Fisher information. Due to the complexity of the computation process for this principle, we provide the derivation in the explanation of Equation (10). Upon careful observation, it can be seen that the two terms in Equation (10) correspond to the empirical risk minimization loss and the structural risk minimization loss, which aligns with Equation (2).
>
> 3.Regarding feature selection and feature space optimization
>
> The method we propose is not a feature selection algorithm, but an evaluator for the feature space selected by feature selection algorithms. Therefore, our method cannot be classified as a feature selection algorithm. As stated in Section 2 of the paper, we use feature selection algorithms as an example to evaluate the effectiveness of the EASE evaluator we propose.
>
> 4.Regarding time complexity
>
> In order to ensure a fair comparison between the traditional evaluators and our proposed method, the traditional evaluators must first be optimized (i.e., parameter tuning). Only after this optimization can we compare them with our method; otherwise, the comparison would not be meaningful. The experiments in Section 5 demonstrate this point.

---

> > ### Comment · Reviewer_WB9F · 2024-11-25
> >
> > Thank you for the response.
> >
> > I still think that the mathematical descriptions in this paper are vague in many aspects (especially because it is unclear what 'space' refers to, which makes it difficult to follow the discussion even). Furthermore, since the authors argue that feature selection is just one application, in order to demonstrate the generality of the method, it would be desirable to validate its effectiveness on tasks other than feature selection. Additionally, to validate the effectiveness of the iterative approach in feature selection, I think it is better to conduct experimental comparisons with embedding-based feature selection methods using neural networks.

---

> > > ### Author Response · Authors · 2024-11-27
> > >
> > > Dear reviewer WB9F：
> > >
> > > Thank you very much for taking the time to review and respond to our comment. This has greatly contributed to improving the quality of our paper.
> > >
> > > 1.Regarding term “space”
> > >
> > > In fact, we are not the first researchers to refer to the matrix formed by combining all features as the "feature space." Many researchers have used this term. I have selected some recent studies as examples, such as: "Over-sampling strategy in feature space for graphs based class-imbalanced bot detection," "Feature selection techniques for machine learning: a survey of more than two decades of research," and "Multi-label feature selection with global and local label correlation." As you can see, the term "space" is used in all of these papers and is widely adopted in this field.
> > >
> > > 2. Regarding effectiveness in tasks beyond feature selection
> > >
> > > Thank you very much for your suggestion. We have added an evaluation of the effectiveness of our proposed evaluator in feature generation methods in the appendix. The comparison algorithm we selected is a DQN-based feature generation algorithm. This paper addresses challenges in representation space reconstruction by proposing a cascading reinforcement learning approach that automates feature generation and selection, enhances explainability, and optimizes feature space reconstruction for machine learning tasks.  Based on the results from the experiment in Appendix C.9, we can see that EASE still demonstrates good robustness in evaluation performance for neural network-based methods.

---

### Official Review · Reviewer_eoma · 2024-10-26

**Soundness:** 3
**Presentation:** 2
**Contribution:** 2
**Rating:** 5
**Confidence:** 3

**Summary:**

This paper designs a new framework (i.e., EASE) to enhance the optimization of feature spaces in machine learning tasks. EASE addresses common limitations in existing methods, such as evaluation bias, poor generalization, and inefficient training. It comprises two main components: the Feature-Sample Subspace Generator and the Contextual Attention Evaluator. The former mitigates evaluation bias by decoupling information within the feature space, while the latter captures evolving patterns incrementally for efficient evaluation. The framework is tested on twelve real-world datasets, demonstrating its effectiveness and efficiency over traditional methods.

**Strengths:**

1. The paper effectively tackles prevalent issues in feature space optimization, such as bias, generalization, and training inefficiency, providing a more robust evaluation methodology.

2. The combination of Feature-Sample Subspace Generator and Contextual Attention Evaluator offers a comprehensive solution that both decouples complex interactions and captures evolving patterns in feature spaces.

3. The extensive experiments on multiple real-world datasets substantiate the superiority of EASE in terms of both accuracy and efficiency, enhancing the paper's credibility.

**Weaknesses:**

1. The proposed method conduct the efficiency comparison experiments of different feature space evaluators across various datasets. It would be better to further analyze the time complexity of different feature space evaluators to demonstrate the efficiency of the proposed method.

2. The proposed method is applied to two iterative feature selection frameworks to validate its effectiveness and generalization capability, i.e., RFE and FLSR. However, both the two baselines are out-of-date. Are there any recent baselines can be applied to?

3. The used several datasets are relatively small, i.e., almost all of their sample sizes are small than 10 thousand. How about applied the proposed method on large-scale datasets?

4. I am a little concerned about the innovativeness of the proposed method, because Section 4.2 contextual attention evaluator seems to simply apply the self-attention mechanism.

**Questions:**

See above.

---

> ### Author Response · Authors · 2024-11-17
>
> Dear reviewer eoma,
>
> Thank you very much for your thorough review of our work and for acknowledging our contributions to feature space optimization. We appreciate your insights regarding the limitations of our paper, and we provide detailed responses below to address these points and improve our work further.
>
> 1.Regarding time complexity
>
> We provide a detailed analysis of the time complexity of each evaluator in Section 5.2.2 of the paper and further supplement this with time complexity experiments under various scenarios in Appendix C.3 and Appendix C.6. The experimental results of the evaluators across different datasets effectively demonstrate the validity of the proposed method.
>
> 2.Regarding the state-of-the-art baseline methods
>
> Our EASE framework is a general framework that can be applied to any algorithm requiring iterative optimization of feature spaces. Feature selection is just one application scenario; moreover, iterative feature generation is another scenario that can be used for validation. Therefore, the core focus of the method is on evaluation. We apologize for not using the state-of-the-art feature space selection algorithms, and we have conducted new experiments on the state-of-the-art feature selection algorithm SDAE at AAAI 2024. On 14 public datasets, our EASE achieved the best performance on 11 datasets and the second-best performance on 3 other datasets (See Appendix C.5). We can observe that in the feature space selected by the SDAE, our proposed EASE exhibits the best evaluation performance in both classification and regression tasks from Appendix C.5. This proves that EASE has stronger robustness.
>
> 3.Regarding large-scale datasets
>
> Thank you for your valuable suggestions. We sincerely appreciate your feedback, which is crucial for improving our work. Based on your suggestion, we have further refined our experimental design and updated the results of EASE on large-scale datasets to provide a more comprehensive and detailed demonstration of its performance (See Section 5.1 for large-scale datasets and Section 5.2 and Appendix C.2, C.3, C.5 for results.). On the large-scale dataset AmazonEA, the EASE evaluator achieved an accuracy of 0.963, precision of 0.782, F1 score of 0.50, and recall of 0.503. On the Nomao dataset, EASE performed with an accuracy of 0.952, precision of 0.947, F1 score of 0.936, and recall of 0.922. Compared to other datasets, EASE demonstrated superior performance. The experimental results on large-scale datasets also validate the effectiveness of the method we proposed.
>
> 4.Regarding Innovation
>
> In fact, the attention mechanism is not the focus of our innovation. Our main focus is on how to address sample bias and construct a general-purpose evaluator with strong generalization performance. The attention mechanism is merely a tool we use to capture important information between two consecutive iterations. In fact, the attention mechanism can be replaced by other feature learning methods, such as CNN for capturing local features. Our innovation lies in proposing a flexible and general framework that allows various feature extraction methods to be integrated, thus optimizing evaluation performance across different tasks.

---

> > ### Author Response · Authors · 2024-11-26
> >
> > Dear Reviewer eoma,
> >
> > Thank you for your valuable feedback on our paper. As the ICLR public discussion phase is ending in a few days, we would like to confirm if our responses have fully addressed your concerns. If there are any remaining issues, we’d be happy to provide further clarifications.
> >
> > If you feel that all concerns have been resolved, we hope this could be reflected in your evaluation.
> >
> > We sincerely appreciate your time and thoughtful input!

---

### Official Review · Reviewer_67BS · 2024-11-02

**Soundness:** 2
**Presentation:** 3
**Contribution:** 2
**Rating:** 5
**Confidence:** 4

**Summary:**

This paper proposes a general adaptive feature space evaluator named EASE, designed to optimize feature spaces through incremental updates and a contextual attention mechanism. By generating feature-sample subspaces and conducting incremental evaluations, this framework improves the efficiency and accuracy of feature selection, with its performance validated across multiple real-world datasets.

**Strengths:**

1. The EASE framework includes two key components: a feature-sample subspace generator and a contextual attention evaluator. The feature-sample subspace generator creates feature subspaces relevant to downstream tasks, allowing the evaluator to focus on the most challenging subspaces in each iteration.
2. An incremental update strategy is introduced in this paper, which retains historical parameter weights and updates only critical parameters when new feature spaces appear, reducing the computational cost of retraining from scratch each time. Additionally, an Elastic Weight Consolidation (EWC) strategy is used to calculate Fisher information.

**Weaknesses:**

1 The paper presents the innovative EASE framework. However, it provides limited implementation details, such as hyperparameter configurations, optimization strategies, and specific training processes.
2 The paper compares EASE with some common models (such as GBDT and Random Forest) but lacks detailed comparisons with current state-of-the-art feature selection or feature space evaluation methods.
3 The theoretical analysis of EASE relies on specific assumptions about data distribution and feature relevance, which may not always align with real-world data, especially in datasets where features are weakly related or uncorrelated.

**Questions:**

1. The complexity of feature subspace generation and multi-head attention mechanisms still makes the EASE framework computationally intensive during training. Due to the large number of parameters in multi-head attention, which require constant updating, the efficiency of EASE may be suboptimal in large-scale datasets or high-dimensional feature spaces.
2. The regularization parameters mentioned in this paper significantly impact the stability and adaptability of parameters in incremental updates, but detailed tuning strategies for different scenarios are lacking. Additionally, the size and sampling strategy of the feature-sample subspaces generated in different iteration steps could directly affect the accuracy and computational cost of final feature space evaluation.
3. While the EASE framework performs well experimentally, it lacks theoretical analysis to explain its generalizability across different feature spaces and tasks. For instance, the paper does not provide detailed theoretical support for whether incremental updates are effective across all feature space distributions, or whether the contextual attention mechanism is universally applicable.

---

> ### Author Response · Authors · 2024-11-17
>
> Dear reviewer 67BS：
>
> We sincerely appreciate the reviewer's time and effort in assessing our paper. We want to answer the posed questions to improve our paper's quality and clarity.
>
> 1.Regarding implementation details
>
> Due to the limitations of the conference page length, we have included all details of the hyperparameter configurations, optimization strategies, specific training processes, and environmental settings for the experiments in Appendix C.1. Please refer to Appendix C.1 for more information. We apologize for not providing an explanation in the main text earlier.
>
> 2.Regarding the state-of-the-art baseline methods
>
> Existing feature space evaluation algorithms typically rely on commonly used models, while research on new evaluators is almost nonexistent. This research gap is why we aim to propose a new feature space evaluator. Besides, we apologize for not using the state-of-the-art feature space selection algorithms. And we additional conducted new experiments on the state-of-the-art feature selection algorithm SDAE at AAAI 2024. On 14 public datasets, our EASE achieved the best performance on most datasets(See Appendix C.5). We can observe that in the feature space selected by the SDAE, our proposed EASE exhibits the best evaluation performance in both classification and regression tasks from Appendix C.5. This proves that EASE has stronger robustness.
>
> 3.Regarding theoretical analysis of EASE
>
> In fact, the EASE method we proposed does not require special attention to the correlations between features and data distribution, as the neural network-based structure can adaptively adjust the usage of features. This is one of the advantages of our method. We have detailed the theoretical aspects of EASE in Section 4.1 and Section 4.3.
>
> 4. Regarding time complexity
>
> In iterative feature space optimization algorithms, traditional evaluation methods often become highly complex when handling large-scale datasets. For effective evaluation, these methods typically need to continuously search for optimal parameters and retrain from scratch on the training dataset during each iteration. Particularly on large-scale or high-dimensional datasets, traditional evaluation algorithms may be unable to yield results due to excessive runtime. We encountered this issue when working with large datasets, which motivated us to propose the EASE framework. In the EASE framework, parameter selection only needs to be done during the initial training phase; in subsequent evaluations, only the parameters of newly emerging features need to be updated, significantly reducing algorithmic complexity. Our experimental results (see Section 5.2.2 and Appendix C.2) confirm this improvement.
>
> 5. Regarding parameter
>
> (1) The regularization parameters have a significant impact on the stability of EASE in different scenarios. Our experimental results (see Section 5.2.1) confirm this effect, as EASE consistently performs best across various datasets.
> (2) As shown in the methodology section of this paper (see Section 4.1), the sampling strategy within the EASE framework is fixed. The loss evaluation for different samples in each iteration effects the sampling steps in the next iteration, so this parameter does not have a major impact on the results.
> (3) To address the issue caused by varying feature space length across different iterations, we set the embedding dimension of the attention mechanism to match the sample size of the feature subspace, effectively resolving the problem. For detailed experiments on the embedding dimension D (the sample size of the feature subspace), please refer to Appendix C.7. We have rephrased this part in the appendix.
> (4) Other parameters: We have also tested the effect of some parameters on the final evaluation results. For details, please refer to Appendix C.7.
>
> 6. Regarding tuning strategies
>
> The details of the tuning strategies used in the experiments are provided in Appendix C.1.
>
> 7. Regarding generalizability
>
> (1) Theoretical Support: We provide detailed theoretical support in Section 4.1 of the methodology. Specifically, in the "Sample Index Optimizer" and "Feature Subspace Construction" subsections, we explain how we leverage limited samples and features to enhance the generalization performance of the evaluator. First, for each iteration of training samples, we apply dynamic weighting to the samples based on the loss from the previous iteration before sampling. Secondly, during feature subspace construction, selection of subspace samples further improves generalization performance.
> (2) Effectiveness: In Section 5.2, we use as many datasets as possible to validate the effectiveness of EASE. The more datasets included, the greater the variety of feature spaces, allowing us to more comprehensively test EASE's performance across different feature spaces. In future work, we will continue to explore the effectiveness of this method on a wider range of datasets.

---

> > ### Author Response · Authors · 2024-11-26
> >
> > Dear Reviewer 67BS,
> >
> > Thank you for your valuable feedback on our paper. As the ICLR public discussion phase is ending in a few days, we would like to confirm if our responses have fully addressed your concerns. If there are any remaining issues, we’d be happy to provide further clarifications.
> >
> > If you feel that all concerns have been resolved, we hope this could be reflected in your evaluation.
> >
> > We sincerely appreciate your time and thoughtful input!

---

> > ### Comment · Reviewer_67BS · 2024-11-27
> >
> > Thank you for your detailed response. While the clarifications provide additional context, several concerns remain that impact the paper's rigor and presentation. The theoretical analysis of EASE, while highlighting its adaptability, lacks sufficient depth to convincingly support its claimed advantages. Discussions on the core strengths and limitations of the method remain underdeveloped. The key implementation details only in Appendix C.1 limits the accessibility of the work, as critical configurations and methods should be summarized in the main text for transparency and replicability. Overall, the paper would benefit from a more robust presentation of results, deeper theoretical insights, and clearer integration of critical details into the main text.

---

> > > ### Author Response · Authors · 2024-11-28
> > >
> > > Dear Reviewer 67BS,
> > >
> > > Thank you for taking the time to read and respond to our comments. Your feedback is crucial for improving the quality of our paper.
> > >
> > > 1.Regarding the key advantages of the proposed method
> > >
> > > Overall, in the  Section 1, we provide a detailed discussion of the limitations of existing methods. In contrast, the proposed method effectively addresses the following shortcomings: （1）Evaluation bias. Existing methods do not account for variability between samples, which limits the evaluator’s ability to capture the full range of features of the space.
> > > (2) Non-generalizability.  Existing methods  need to evaluate the feature space by replacing the evaluator based on specific requirements limits its ability to capture generalizable patterns. (3) Training Inefficiency. Existing methods need to  retrain the evaluator from scratch at each iteration significantly increases computational demands.
> > >
> > > In detail, Section 4 provides a comprehensive explanation of the specific design used to address the corresponding limitations. Specifically, (1) The FEATURE-SAMPLE SUBSPACE GENERATOR can address sample bias by decoupling the feature space (see Section 4.1); (2) CONTEXTUAL ATTENTION EVALUATOR can facilitate comprehensive information extraction (see section 4.2) ; (3) OPTIMIZATION  can  incrementally update the parameters of the contextual attention evaluator, enabling faster updates and accelerating the entire feature space optimization process (see section 4.3). These points represent the key advantages of the method we proposed.
> > >
> > > 2.Regarding the Limitations of the Proposed Method
> > >
> > > In Section 6, we explicitly highlight the limitation of our method, namely that it does not currently account for distribution shift. This is a key issue we plan to address in our future work.
> > >
> > > 3.Regarding Experimental Setup
> > >
> > > In response to your comments, we have provided additional explanations of our experimental setup in the main text. We have outlined the relevant details in three parts:
> > > First, in the main text, we primarily discuss the methods used and the experimental results; Second, in the appendix, we provide detailed descriptions of the parameters used and some training tricks; Finally, we have made the code open-source, which includes all the implementation details of EASE within the feature selection framework to facilitate reproducibility and further research.

---

### Meta-Review · Area_Chair_c6zH · 2024-12-18

**Metareview:**

Thanks for your submission to ICLR.

This paper received borderline reviews, leaning to reject.  All three reviewers noted several issues with the manuscript, including i) limited implementation details / issues with presentation, ii) missing empirical comparisons (data sets, baselines), iii) concerns about the theoretical analysis, iv) novelty.  The rebuttal addressed some of these issues, but during the discussion phase, two of the three reviewers responded that they still felt the paper was not ready for publication.  The third reviewer did not respond.

Given the issues that persist after the discussion phase, and the relatively low scores, I think this paper needs some additional work before it is ready for publication.  Please keep in mind the comments of the reviewers when preparing a future version of the paper.

**Additional Comments On Reviewer Discussion:**

Two of the three reviewers responded, and ultimately kept their scores as they were prior to the discussion phase.

---

### Decision · Program_Chairs · 2025-01-22

Reject